# Mapping the drivers of within-host pathogen evolution using massive data sets

Duncan S. Palmer [1,2,3], Isaac Turner [1,2], Sarah Fidler[4], John Frater[3,5,6], Dominique Goedhals[7], Philip Goulder[8,9], Kuan-Hsiang Gary Huang [5,10], Annette Oxenius [11], Rodney Phillips[3,5,6,12], Roger Shapiro[13,14], Cloete van Vuuren[7], Angela R. McLean[3,15] & Gil McVean [1,2,16]

Differences among hosts, resulting from genetic variation in the immune system or heterogeneity in drug treatment, can impact within-host pathogen evolution. Genetic association studies can potentially identify such interactions. However, extensive and correlated genetic population structure in hosts and pathogens presents a substantial risk of confounding analyses. Moreover, the multiple testing burden of interaction scanning can potentially limit power. We present a Bayesian approach for detecting host influences on pathogen evolution that exploits vast existing data sets of pathogen diversity to improve power and control for stratification. The approach models key processes, including recombination and selection, and identifies regions of the pathogen genome affected by host factors. Our simulations and empirical analysis of drug-induced selection on the HIV-1 genome show that the method recovers known associations and has superior precision-recall characteristics compared to other approaches. We build a high-resolution map of HLA-induced selection in the HIV-1 genome, identifying novel epitope-allele combinations.

[1] Department of Statistics, University of Oxford, 24-29 St Giles', Oxford OX1 3LB, UK. [2] Wellcome Trust Centre for Human Genetics, Roosevelt Drive, Oxford OX3 7BN, UK. [3] Institute for Emerging Infections, The Oxford Martin School, Oxford OX1 3BD, UK. [4] Division of Medicine, Wright Fleming Institute, Imperial College, London W2 1PG, UK. [5] Nuffield Department of Clinical Medicine, University of Oxford, Peter Medawar Building for Pathogen Research, Oxford OX1 3SY, UK. [6] Oxford NIHR Biomedical Research Centre, Oxford OX3 7LE, UK. [7] HIV Pathogenesis Programme, Doris Duke Medical Research Institute, University of KwaZulu-Natal, Durban 4013, South Africa. [8] Division of Infectious Diseases, University of the Free State, and 3 Military Hospital, Bloemfontein 9300, South Africa. [9] Department of Paediatrics, University of Oxford, Peter Medawar Building for Pathogen Research, Oxford OX1 3SY, UK. [10] Einstein Medical Center Philadelphia, 5501 Old York Road, PA 19141, USA. [11] Institute of Microbiology, Swiss Federal Institute of Technology Zurich, 8093 Zurich, Switzerland. [12] Faculty of Medicine, UNSW Sydney, NSW 2052, Australia. [13] Botswana Harvard AIDS Institute Partnership, Gaborone BO 320, Botswana. [14] Department of Immunology and Infectious Diseases, Harvard TH Chan School of Public Health, Boston, MA 02215, USA. [15] Zoology Department, University of Oxford, South Parks Road, Oxford OX1 3PS, UK. [16] Big Data Institute, Li Ka Shing Centre for Health Information and Discovery, University of Oxford, Old Road Campus, Oxford OX3 7LF, UK. Correspondence and requests for materials should be addressed to D.S.P. (email: duncan.stuart. palmer@gmail.com)

Variation in multiple host factors, both genetic and non-genetic, can influence the genetic composition of infecting pathogens and their subsequent evolutionary trajectory within a host. Examples, include human leukocyte antigen (HLA) restriction of epitopes and subsequent escape in HIV-1 and other viruses[1–6], drug-induced selection pressure and appearance of drug-resistance mutations in viruses, bacteria and eukaryotic pathogens[7–13] and interactions between polymorphic red-blood cell types and malarial disease[14,15]. Consequently, the molecular mechanisms underlying diverse pathogen-related processes including infection, invasion, immune-response and drug resistance, can potentially be uncovered by studying the association between host factors and the genetic composition of pathogens[16–22].

However, while it is now feasible to collect large-scale data on pathogen genomic variation and host parameters, reliable hypothesis-free detection of biologically meaningful associations between host factors and pathogen diversity is challenging for several reasons; the greatest of which is population structure. Host genetic variation often has a strong spatial structure arising from historical patterns of isolation and gene flow. Because of their commensal nature and mode of transmission, most pathogens are likely to share some of this structure, leading to non-causal association between host and pathogen genetic variation. For the same reason, geographical heterogeneity among host factors that influence pathogens causally (e.g., local variation in treatment protocols) may also lead to indirect correlation between host and pathogen genetics.

A second major challenge is statistical power. Consider searching genome-wide for associations between host genomic factors and pathogen genomic factors, in which the number of tests carried out could be in the billions. Naïve correction for multiple testing is likely to eliminate power for anything except the strongest associations. Consequently, there is a need for approaches to association testing that enable prior information about the likely structure of association to be used in the search for signal.

To date, various approaches to testing for association between pathogen genetic variation and host factors, both genetic, as in the case of classical HLA loci, and non-genetic, as in the case of drug resistance, have been developed[16–18,23,24]. Standard association tests, which suffered the problems of stratification described above, were superseded by methods that utilise an inferred phylogeny of pathogen samples to correct for relatedness[17,22,25–28]. Moreover, additional power can be obtained by explicit modelling of the processes of escape and reversion in the

context of HLA restriction of HIV-1[17,22,23]. However, such approaches have a number of limitations. For example, they do not consider recombination in pathogen genomes, they do not make use of all the data available, they typically do not infer the strength of host-induced selection or combine information across nearby sites in the context of epitope mapping and the more sophisticated approaches are often computationally prohibitive for very large samples.

To address these limitations we have developed a model-based approach to inferring the effect of host factors on pathogen genomic variation. The approach is motivated by the presence of extremely large databases of pathogen genomic data. Given the size of the databases (e.g., the Los Alamos database on HIV-1 has over 150,000 sequences encoding a portion of reverse transcriptase), it is likely that there are sequences closely related to those that infected the individuals within a particular study in question. Our approach aims to infer the most likely ancestral infecting sequence (which may be a mosaic of those in the database) for each individual and therefore to identify the pathogen evolution that has resulted directly from exposure to the current host. Moreover, we model correlations in evolution among adjacent sites within the pathogen that arise through being a shared target of selection (e.g., within a restricted HLA epitope or within a region of a protein where drug resistance and consequent compensatory mutations can evolve). A diagram of the underlying process and our modelling framework is shown in Fig. 1. Our methods and inference regime are detailed in the Methods and Supplementary Methods. We show that the approach has substantially improved power to detect sites under selection and, through applications to the evolution of drug resistance and escape from HLA-drive immune-restriction in HIV-1, how the method can deliver new insight into important biological processes.

## Results

**Simulations and methods comparison.** To validate the method and compare its performance to alternative approaches for detecting host-induced selection we carried out three simulation studies. First, we simulated data under the fitted model to evaluate power and accuracy. We used HIV-1 *protease* sequence data from three major public databases as a reference (Table 1; $n = 162,901$), and simulated 100 replicates of a data set of 460 study sequences with six different HLA-induced selection profiles, four of which were common ($n = 100$ each) and two of which were rare ($n = 30$ each); Supplementary Fig. 1. We

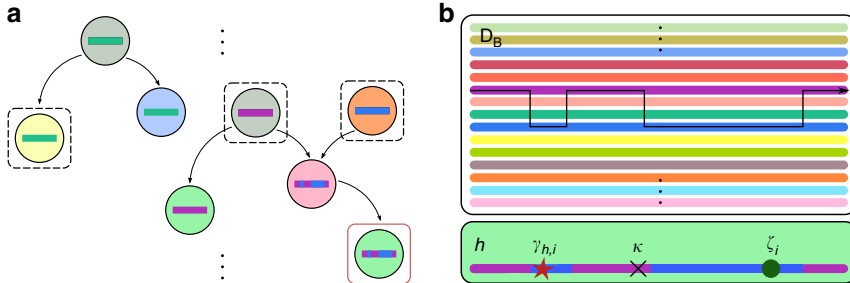

**Fig. 1** The underlying process and our approximation. **a** The underlying process: infected hosts are circles, coloured by host factor information. Coloured strips represent the consensus pathogen sequence within infected hosts, and arrows indicate the direction of infection. Sampled hosts with pathogen sequence information are outlined. A red outline indicates that host factor information is also available. **b** Our approximation: pathogen sequences in **D** are generated from $\mathbf{D}_B$, modulated by host factors of **D**. As $|\mathbf{D}_B| \gg 0$, we assume $\mathbf{D}_B$ is the set of all possible sequences an individual can be infected with. Members of **D** arise from $\mathbf{D}_B$ through recombination and mutation. We assume all selection along a lineage connecting each member of **D** to its closest neighbour in $\mathbf{D}_B$ (subject to recombination) occurred within the host that the member of **D** was isolated from. Thus only host factor information for **D** is required. In **b**, there is one green host factor, $h$. Recombination, shown by the arrow, results in the colouring of the sequence. Three mutations have occurred: one due to $h$-associated selection (red star), one reversion (green circle), and one synonymous transition (black cross)

**Table 1 Data used in this study: public databases**

| Database | Cohort size | Details |
|---|---|---|
| Los Alamos HIV Sequence Database[54] | Protease n = 119,878<br>Reverse transcriptase n = 148,866 | Fed by biweekly downloads of new HIV sequence data submitted to GenBank. Metadata is extracted from GenBank submissions and manually annotated with additional information taken from the corresponding publications and through direct interaction with authors[55]. |
| Stanford Drug Resistance Database[9] | Protease n = 81,533<br>Reverse transcriptase n = 88,780 | Collection of protease and reverse transcriptase sequences with associated patient drug prescription data. |
| HIV positive selection database[56] | Protease n = 45,161<br>Reverse transcriptase n = 45,161 | Protease and reverse transcriptase sequence data taken from HIV-1 patient plasma samples by Specialty Laboratories from 1999 to 2002[56]. The database was created in order to identify regions of drug selection using estimates of non-synonymous/synonymous base changes along the viral genome. |

performed simulations with recombination ranging between 0 and 0.01 (a rate of 0.01 meaning that on average the ancestor of each study sample copies 100 contiguous codons from a reference sample before recombination). We performed joint inference on the HLA-associated selection parameters across the viral sequence and examined how the size of the sample-specific reference panel (range: 10–100) affects estimates. Motivation behind the use of sample-specific reference panel and our approach is given in the Methods subsection: Restricting $D_B$: sample-specific reference data sets. In the absence of recombination, parameter estimates are largely unbiased and accurate, although the rate of reversion has high variance (Fig. 2; Supplementary Fig. 2). Estimates for rare alleles perform similarly to those for common alleles (Fig. 2b). We find little impact of the size of the sample-specific reference data set and the posterior distributions are well calibrated (Supplementary Fig. 3). Recombination leads to a downward bias in estimates of recombination and selection intensity and upward bias in the rate of reversion (Fig. 2c; Supplementary Fig. 4) and consequently poorer posterior calibration at sites under selection (Supplementary Fig. 4). Nevertheless, the inferred profiles of host-dependent and host-independent selection pressures remain strongly correlated to the truth. We also considered the impact of error in choosing the sample-specific reference panel by forcing inclusion of the sequences actually copied. We find that forcing the true ancestors to be included improves accuracy by only a small margin (Supplementary Fig. 5) indicating that 100 potential ancestors chosen through Hamming distance is typically sufficient.

In the second set of simulations we assessed robustness by simulating data under a birth–death model (without recombination), setting the current infected population size at 1,000,000 and sampling proportion at 10% to define $D_B$ and sample 1,460 query sequences with associated host HLA information (see Methods subsection: Simulation study 2: simulating a sampled birth–death process for details). We find some attenuation of the selection signal (and over-estimation of the reversion parameter) but strong correlation between the inferred and estimated strengths of host-dependent and host-independent selection (Fig. 2c; Supplementary Fig. 6).

The birth–death simulations also enable comparison of our approach with five alternative methods for identifying sites under host factor specific selection (Supplementary Methods subsection: Simulation study 2: methods comparison): Fisher's exact test (as in Moore et al.[16]); a phylogenetically corrected Fisher's exact test (as in Bhattacharya et al.[22]); an approximate escape rate estimate (as in Fryer et al.[18]); a 'Phylogenetic dependency networks' approach—PhyloD (as in Carlson et al.[25]); and PhyloD OR (as in Carlson et al.[21]). In each case, we assume that the wild-type consensus strain is correctly defined and set any non-synonymous difference from consensus as a candidate for escape. For each method, we obtained p values or parameter estimates

that provide a metric for the strength of selection conditional on each HLA type and generated receiver operating characteristic curves for each simulation (Fig. 2e). We find that our approach dramatically increases sensitivity for a given false-positive rate (FPR). For example, at FPR = 0.01, our sensitivity is 0.61, compared to the second best-performing method, PhyloD, at 0.13. To assess whether the difference between methods decreases with sample size, we repeated the analysis with 3000 query sequences. Our method achieves a sensitivity of 0.81 (at an FPR of 0.01), compared to the next best-performing method, PhyloD, at 0.22 (Supplementary Fig. 7). We conclude that augmenting study data with a large reference data set and modelling the escape process explicitly provides a substantial gain in the ability to identify sites under host factor specific selection.

To examine the impact of a reduced reference sequence data set, we randomly subsampled 10 and 1% of $D_B$ from Simulation study 2. We also considered a leave-one-out (LOO) strategy, where the reference data are augmented with study sequences, with the exception of the sample under consideration. We find that larger reference panels achieve greater accuracy. For example, at FPR = 0.01, sensitivity ranges from 0.61 with the full reference to 0.49 with 10% and 0.32 with 1%. The LOO strategy only boosts power if the reference panel is of the order of the sample size. However, in the absence of any reference panel, our method with a LOO strategy still achieves considerably greater power (sensitivity of 0.34 at FPR = 0.01) than any competing method (Supplementary Fig. 8).

In the third simulation study (Methods subsection: Simulation study 3: the effect of population differentiation), we used an empirical bootstrap approach to assess robustness to population structure and the impact of both differentiation between $D$ and $D_B$ and relatedness within $D$, taking the parameters estimated from empirical data (see Results: HLA-associated selection). We used viral sequence data from Botswana (n = 343, protease only) to simulate ancestors and the empirical distribution of HLA genotype frequencies from this sample. We measured the accuracy of estimates (both in terms of bias and calibration of posteriors) for different protease reference panels, ranging from the ideal panel (i.e., the ancestors used in the simulation), to that used in this study, to one that considered only the 57,969 sequences known not to come from Botswana. The latter panel maximises differentiation and also introduces the potential problem of sequences within $D$ typically being more closely related to each other than to members of $D_B$. Results are summarised in Supplementary Figs. 9–11. In terms of bias (summarised by the frequency-weighted root mean square error (RMSE) between true and inferred HLA-associated selection profiles) we find relatively little differences between panels, with the most diverged reference panel being only 3% worse than the best. The estimator was well calibrated for each reference panel used: 95% and 50% credible intervals contained the true value at

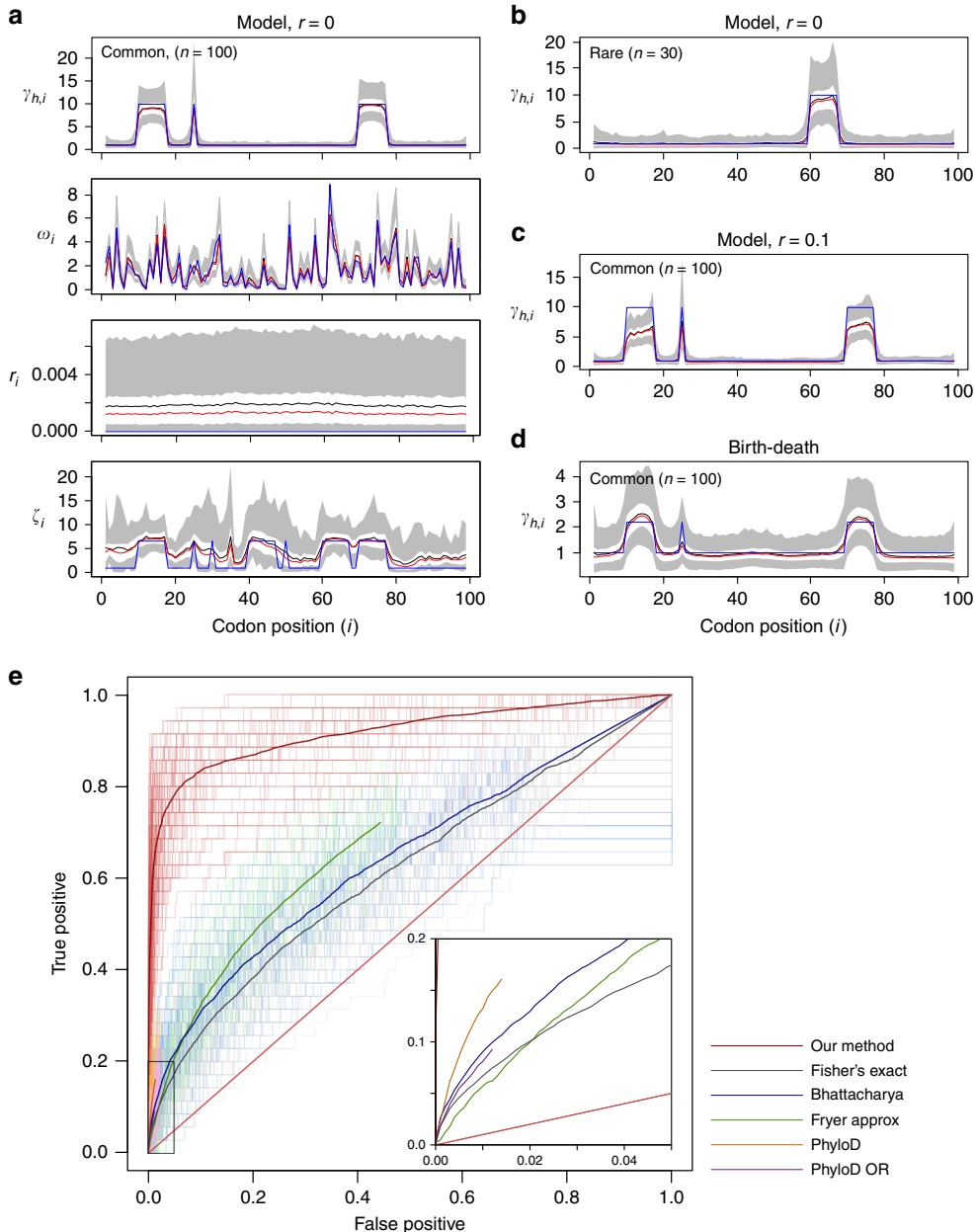

**Fig. 2** Simulation results summary. Inference results for Simulation studies 1 and 2 are shown in **a–d**. **a** Simulation study 1; $r = 0$. $\gamma$ for a common HLA ($n = 100$), $\frac{dN}{dS}$ ratio, recombination probability between adjacent sites ($r$), and reversion scaling ($\zeta$). **b** Simulation study 1; $r = 0$. $\gamma$ for a rare HLA ($n = 30$). **c** Simulation study 1; $r = 0.01$, $\gamma$ for a common HLA ($n = 100$). **d** Simulation study 2, $\gamma$ for a common HLA ($n = 100$). In **a–d**, (except $\omega$ in **a**), averages are taken over 100 independent MCMC runs on independent simulated data with the same underlying parameters. The true value is shown in blue, mean and median estimates are in black and red, respectively. White bands enclose 50% credible intervals, in turn enclosed by grey 95% credible intervals. In **c**, the truth is rescaled by the average number of individuals between a randomly chosen pair of leaves in the sampled birth–death tree. For $\omega$ in **a**, averages are taken over a single-MCMC run, chosen at random as $\omega$ differs across simulation runs (sampled from the prior). See Supplementary Figs. 2–5 for full results summaries. **e** ROC curves for existing method. Six methods used to identify HLA-associated selection on viral sequence are applied to data simulated under the birth–death process used in simulation study 2. The Inset zooms in to the region enclosed by the black box. ROC curves for 100 independent birth–death simulations are lightly coloured, and averaged to generate the heavier lines. ROC curves for Fryer Approx, PhyloD and PhyloD OR do not extend to (1, 1). For Fryer Approx this is because we stop our threshold for estimated rates at 0. For PhyloD and PhyloD OR, it is because many sites will not be included in leaf distribution or logistic regression respectively. Source data are provided as a Source Data file

between 91–98% and 45–57% of sites across bootstrap replicates, respectively. Typically, the LOO strategy had little effect on estimator performance. We conclude that the approach gives substantial robustness to population structure and within-sample relatedness, enabling the integration of highly diverse data sets, though note that very highly diverged reference data sets (for example, consisting of a separate subtype) will perform poorly.

**Drug-associated selection.** To provide empirical validation of our approach for detecting host factor dependent selection we analysed HIV-1 data from the Stanford drug resistance database[9] in which viral sequences are linked to antiretroviral drug treatment history of the patient. We therefore aim to learn drug-treatment specific evolution, while other selective factors (e.g., selection due to cytotoxic-T-lymphocyte (CTL) pressure, the

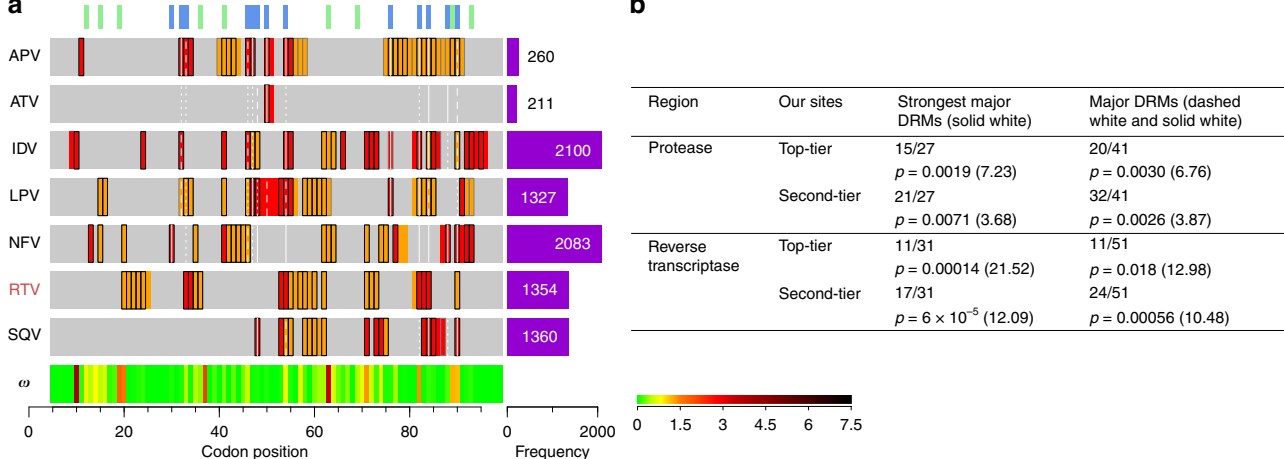

**Fig. 3** Drug-associated selection analysis. **a** Protease results summary. Grey bars highlight the region analysed. Codon position is measured relative to the HXB2 from the start of *protease*. Rows summarise drug-associated selection—drugs are shown to the left. Purple bars show the number of individuals prescribed the drug at viral sampling. Red and orange indicate our median estimate is >2 and >1.5, respectively. Sites are outlined in Black and grey if in addition the 2.5% and 10% quantile is >1, respectively. Green and blue lines at the top of the plots highlight differences between subtype B/C at the amino-acid level, and sites of DRMs[9], respectively. Classes of DRM are displayed: solid white lines tag sites of DRMs which confer the highest levels of resistance for that drug, dashed white lines tag major DRMs[9], and dotted white lines tag DRMs when combined with other mutations[9]. Drugs highlighted red do not have major DRMs in the Stanford drug resistance database. Median $\omega$ is displayed at the foot of the figure according to the colour bar. APV Amprenavir, ATV Atazanavir, IDV Indinavir, LPV Lopinavir, NFV Nelfinavir, RTV Ritonavir, SQV Saquinavir. **b** Sensitivity of inference. Proportions of the strongest major DRMs and major DRMs we identify when using our top-tier and second-tier candidate sites are provided, with associated *p* values and odds-ratios shown in parentheses. Source data are provided as a Source Data file

antibody response, or the APOBEC3G response) will be captured by $\omega$. We analysed *protease* and *reverse transcriptase* independently and excluded integrase due to lack of data[9]. Priors on parameters are given in Supplementary Table 1. We defined the collection of sequences in hosts not receiving any therapy at the time of sequencing as $\mathbf{D}_B$, and viral sequences in hosts receiving any treatment (coupled with their drug regime data) as $\mathbf{D}$. To guarantee the presence of variation in drug selection we introduce a null class by randomly assigning 2,000 sequences from $\mathbf{D}_B$ to $\mathbf{D}$. Details of data preparation are given in the Supplementary Methods subsection: Query and reference data set preparation.

We estimated $\gamma_{h,i}$ for protease and reverse transcriptase inhibitors prescribed to at least 10 individuals in $\mathbf{D}$ (Fig. 3a; Supplementary Fig. 12), and identified sites with very strong evidence for drug-induced escape (median estimate of selection factor >2 and at least 97.5% of the posterior >1; top-tier) and moderate evidence (median estimate of selection factor >1.5 and at least 90% of the posterior >1; second-tier). We compared the enrichment of sites identified to known (experimentally validated) major drug resistance mutations (DRMs)[9] (Fig. 3b). For some drugs, DRM data was lacking because the drug is no longer commonly used (DDC and DLV), was an experimental drug that failed ($\alpha$APA, and ADV) or is now used in small amounts with other drugs (RTV). We observed strong, statistically significant and consistent enrichment of DRMs at sites identified as selected; with elevated enrichment (as measured by odds ratio) in the strongest DRMs and sites with strongest evidence for selection. For example, of the 31 strongest DRMs in reverse transcriptase, 11 are found in the top-tier selected sites and a further 6 in the second-tier. Of the 41 apparent false positive sites, 17 are described as being selected for in the literature (but not classified as a major DRM in the Stanford database); for example mutation at codon 11 of *reverse transcriptase* is known to be associated with minor reductions in APV susceptibility[29], but is not classified as a major DRM[9]. A further three apparent false positive are DRMs for treatment cocktails including the drug, nine are sites of DRMs for drugs in the same class but where more specific

information is not provided, seven are DRMs for different drugs in the same class and only five appear to have no support in the literature (Supplementary Table 2). False negatives are caused by sites where the selected codon is more than one nucleotide change away from the consensus (five mutations), one is an insertion, eight have in vitro support for drug resistance but are not documented as being selected for in vivo, seven have no compelling evidence in the literature and three are genuinely missed (Supplementary Table 3)[9]. Further evidence for the biological validity of the inferred profiles is given by the co-clustering of related drugs in trees constructed from the selection intensity profiles (Supplementary Fig. 13). In summary, we find that the method can identify sites selected by specific drug treatments in vivo, either directly through resistance or indirectly through compensation for resistance mutations that reduce fitness.

**HLA-associated selection.** While the extent to which host HLA alleles can influence patterns of escape and reversion in HIV-1 has been studied extensively, the methods developed here provide additional power and accuracy as well as provide control against factors such as population stratification and recombination. Moreover, the methody also provides a framework in which to combine data from multiple previous studies. We have assembled nearly 3000 HIV-1 sequences from patients with known HLA class I genotypes from six studies, of European and African ancestry representing a mixture of subtype B and C sequences (Table 2). We augmented the data with HIV-1 genome data taken from three sources, representing 250,000–280,000 sequences depending on the gene (Table 1). We analysed the data in two ways, first by considering escape as deviation from subtype B and also by considering escape from subtype C. We expect these analyses to yield similar results, except at sites where subtypes differ systematically. We analysed data from *protease* and *reverse transcriptase* and inferred the impact of alleles at HLA class I loci (at 2-digit resolution) jointly. We restricted analysis to HLA

**Table 2 Data used in this study: viral sequence data with associated host HLA information**

| Data set | Cohort size | Sampling date | Geographic region | Treatment | Study requirements |
|---|---|---|---|---|---|
| SSITT[57,58] (Swiss portion) | $n = 79$ | 2000 | Various cities across Switzerland | HAART | Undetectable VL for >6 months, CD4+ count >300 μl⁻¹. No history of non-nucleoside reverse transcriptase inhibitors. |
| SPARTAC[59] (UK portion) | $n = 258$ | Aug 2003–July 2007 | UK | ART naïve on recruitment | Primary infection, though definition of this is complex, see[59] for details. |
| Bloemfontein[48,60] | $n = 278$ | Feb 2006–Sept 2006 | Bloemfontein, South Africa | ART naïve | Chronic infection, low and high CD4+ count favoured: 96 high (>500 μl⁻¹), 18 medium (200–400 μl⁻¹) and 164 low (<100 μl⁻¹). |
| Durban[61,62] | $n = 1,218$ | 1999–2006 | Durban, South Africa | ART naïve | Recruited following voluntary counselling and testing in antenatal or outpatient clinics. |
| Mma Bana[63] | $n = 514$ | July 2006–May 2008 | Gaborone, Botswana | ART naïve | Pregnant women. |
| Los Alamos | Protease $n = 432$ Reverse transcriptase $n = 334$ | Many | Many | Many | Many |

All remaining sequences with associated host HLA data available in the Los Alamos HIV sequence database[54], which were not present in the above studies constitute those data in the final row

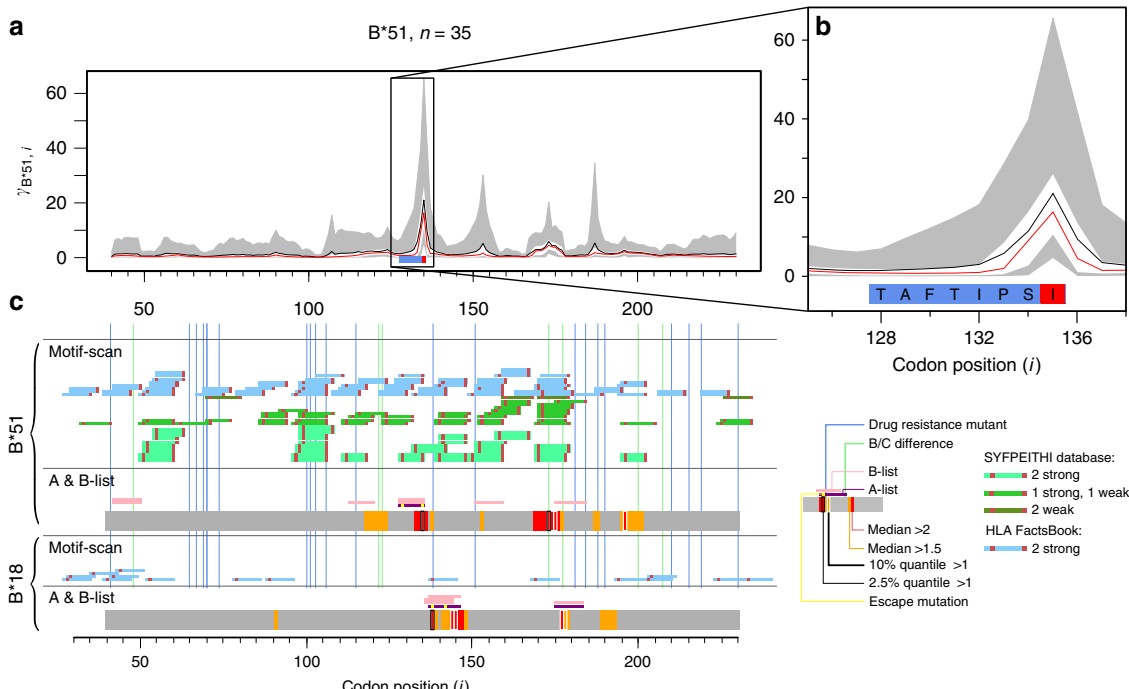

**Fig. 4** Selected results summaries. **a** B*51 associated selection away from $\mathcal{C} = C$ across the analysed region of *reverse transcriptase*. Site numbering is relative to the HXB2, starting from beginning of *reverse transcriptase*. Plotting is as in Fig. 2a–d. The B*51 epitope TAFTIPSI is highlighted by the blue box—the site of the known escape variant is highlighted red. The black rectangle is zoomed in on in **b**. **c** Epitopes predicted by Motif Scan and A-list/B-list epitopes for two example HLA types are displayed. Corresponding results from our inference are shown on the grey strips—colouring is summarised by the key. Source data are provided as a Source Data file

alleles with at least 10 representatives in the data. We truncated to 2-digit HLA resolution to boost power as the majority of alleles are rare (61% of 4-digit HLA alleles have <10 copies, compared to 30% at 2-digit resolution). Further, 27.3% of the individuals for whom we have HLA information were typed to the 2-digit level. Full details of data preparation are given in the Supplementary Methods subsection: Query and reference data set preparation. We define 'top-tier' HLA-associated candidate sites as those where the median $\gamma_{h,i} > 2$ and the lower 2.5% quantile > 1, and 'second-tier' candidate sites as those where the 10% quantile of $\gamma_{h,i} > 1$.

To illustrate the value of our approach, we first considered the B*51 restricted epitope TAFTIPSI in reverse transcriptase

(Fig. 4). We find a strong signal of selection within the epitope, with the escape site (position 135) experiencing the strongest rate elevation. Interestingly, a variant at site 129 which is known to abrogate CTL recognition in vitro[30], is highly conserved in vivo and shows no signal for selection, presumably due to other fitness consequences. In contrast, we also see strong evidence for B*51 associated escape around codons 173 and 195, which have not been reported previously. Other known allele–epitope combinations that we recover include, for example, B*18 associated selection around codon 138[31–33].

To compare to experimental and previous work on epitope restriction we classify documented epitopes into an A-list, which

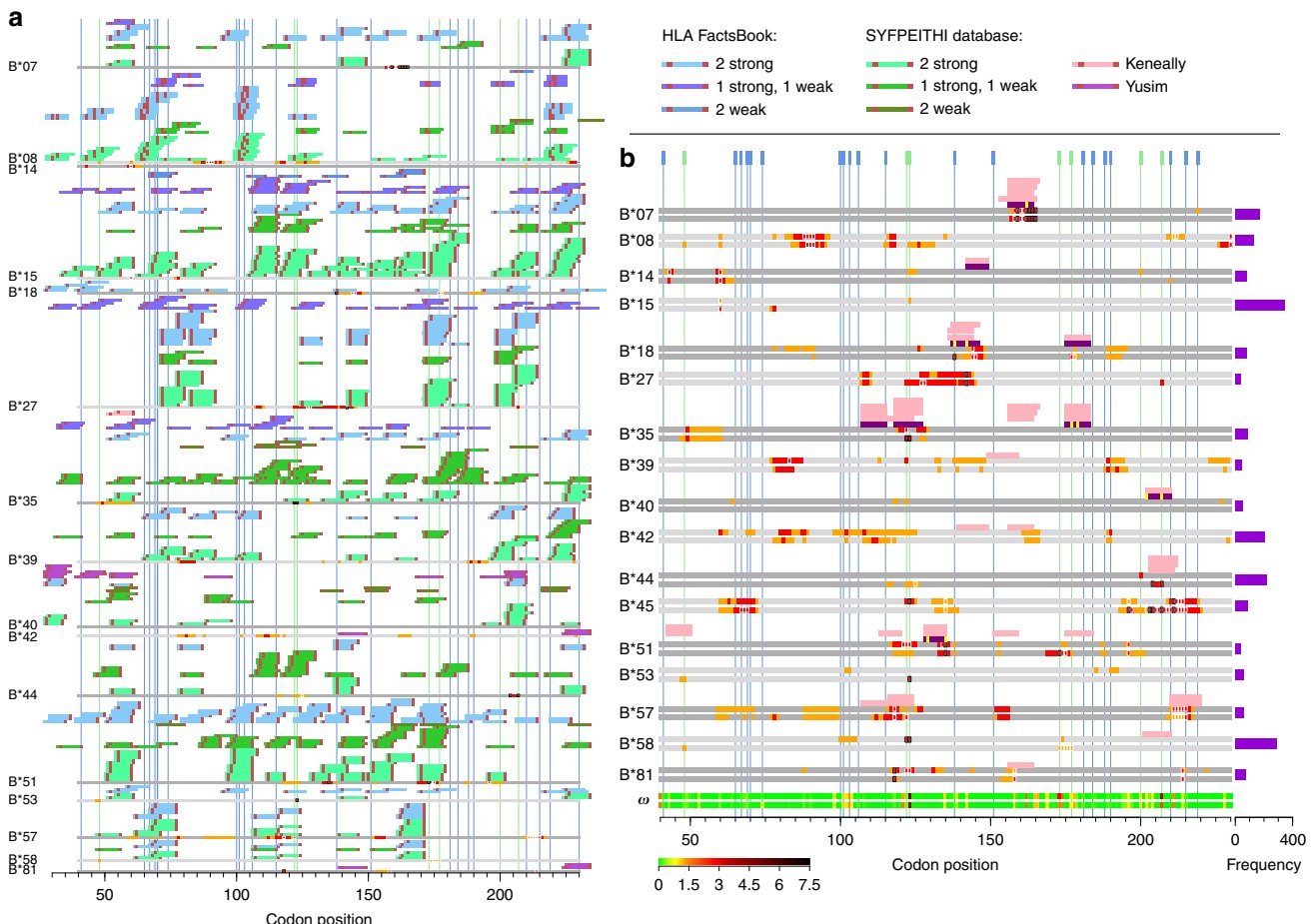

**Fig. 5** HLA-B-associated selection in reverse transcriptase. **a** Motif scan compared to our inference. Inference of HLA-associated selection from $\mathcal{C} = C$ is summarised on the grey strips: sites with median $\gamma_{h,i} > 2$ and 1.5 are coloured red and orange respectively, sites with 2.5% and 10% quantile > 1 are outlined in black and white, respectively. Blue and green vertical lines tag sites of drug resistance mutations and sites that differentiate subtype B/C viruses at the amino-acid level. We scan reverse transcriptase for putative epitopes matching to known binding motifs in the literature; see key. Anchor residues are highlighted in red. **b** A-list/B-list compared to our inference. Inference results on the grey strips are coloured as in **a**. Two grey strips per HLA show selection away from $\mathcal{C} = B$ and $\mathcal{C} = C$ respectively. Bar plots to the right show HLA frequencies. The two lowest strips summarise median $\omega$ according to the colour bar. A-list[34] and B-list[35] epitopes above the grey are coloured purple and pink respectively. Sites of known escape variants within the A-list epitopes are highlighted yellow. Source data are provided as a Source Data file.

represents the best-defined experimentally determined HIV-1 CTL epitopes, updated yearly[34] and a B-list, which refers to the entire collection of epitopes reported in the literature[35]. Among the A-list epitopes, some have documented CTL escape variants, either in vivo or in vitro. See Supplementary Tables 4 and 5 for details and Supplementary Notes. We also consider an in silico set of predictions for strongly-binding anchor residues generated by Motif-Scan[36]. We compare these sites to our estimates of HLA-associated selection across protease and reverse transcriptase (Fig. 5, Supplementary Figs. 9–13).

To assess overlap between sites identified we measured the enrichment of sites identified as under selection and used a permutation strategy to assess significance (Methods subsection: Testing overlap with A-list and B-list epitopes). Across protease and reverse transcriptase we find strong enrichment of top-tier signals of selection at A-list epitopes for HLA-B (Table 3). For example, top-tier sites have an odds ratio of 63 for enrichment in A-list epitopes ($p = 0.003$). As the confidence in the epitope collection or the strength of selection decreases, so does the enrichment. We find no evidence for enrichment of selection at computationally predicted epitopes and little evidence for overlap of HLA-A associated sites, though in general we note that HLA-A alleles show much weaker evidence for selection in general (Supplementary Figs. 14 and 17).

We also find very little evidence for HLA-C driven selection at all (Supplementary Figs. 16 and 18).

Comparing our results to those in Carlson et al.[37], focusing on the well-studied HLA-B associations, we find that 100% of the sites that are both in our top-tier and $q < 0.2$ in Carlson et al.[37] reside in A-list or B-list epitopes (Supplementary Tables 6 and 7). Subsets of method specific HLA associations also lie in A-list and B-list epitopes, which, given the overlap in subtype C viral samples (100% of subtype C viral sequence data with associated host HLA information was present in Carlson et al.[37]) suggest that the approaches are complementary, each likely detecting distinct portions of the true underlying signal. For example, our notion of escape means that we cannot detect HLA-associated selection to the consensus codon.

Overall, only a small number of sites identified here as being HLA-associated have been described and experimentally validated previously; four in reverse transcriptase (Table 4), and one in protease (Table 5). Conversely, not all previously reported escape mutation sites are identified here. In some cases this may be due to low-sample size in this study. However, it is also possible that earlier studies failed to account adequately for linkage disequilibrium between HLA alleles. We also identified potentially novel signals of HLA-driven epitope escape. For

**Table 3 Overlap between sites under selection and known HLA epitopes**

| Region | HLA | Selected sites | Motif scan[a] | B-list | A-list |
|---|---|---|---|---|---|
| Protease | A | Top-tier | 0.693 (0.43) | 1 (0) | 1 (0) |
| | | Second-tier | 0.0855 (2.52) | 0.160 (4.12) | 0.166 (9.00) |
| | B | Top-tier | 1 (0) | 0.0420 (48.8) | 0.0499 (∞) |
| | | Second-tier | 0.964 (0.20) | 0.213 (3.50) | 0.0335 (14.0) |
| Reverse transcriptase | A | Top-tier | 0.953 (0.28) | 0.404 (1.28) | 1 (0) |
| | | Second-tier | 0.935 (0.67) | 0.122 (1.57) | 0.499 (1.61) |
| | B | Top-tier | 0.137 (3.26) | 0.0000290 (55.2) | 0.00278 (63.3) |
| | | Second-tier | 0.303 (1.50) | 0.000250 (19.4) | 0.00278 (37.7) |

Permutation p values for overlap are displayed for each collection of putative epitopes, with odds ratios displayed in parentheses. [a]Putative epitopes were generated using the subtype C consensus; results using the subtype B consensus were similar. Source data are provided as a Source Data file

**Table 4 Codons in *reverse transcriptase* showing evidence for HLA-associated selection**

| HLA | $\mathcal{C} = $ B | $\mathcal{C} = $ C |
|---|---|---|
| A*03 | 135, 166† | 135, 136 |
| A*11 | | 126 |
| A*43 | 118, 135, 139, 140, 141, 142 | 118, 122*, 123*, 135 |
| A*66 | 135 | 135, <u>138</u> |
| B*07 | 159, 162†, 163, 164, 165 | 159, 162†, 163, 164, 165 |
| B*18 | | 138† |
| B*27 | 142 | 142 |
| B*35 | | 122*, 123* |
| B*44 | | 204, 207* |
| B*45 | 123*, 211 | 196, 203, 204, 207*, 211 |
| B*51 | 135† | 135†, 173* |
| B*53 | | 123* |
| B*58 | 122*, 123* | |
| B*81 | 118 | 118 |
| C*04 | <u>210</u>, 211 | |

Sites defined by $\gamma_{h,i} > 2$ and lower 2.5% quantile > 1. Underline: Site of known drug resistance mutations. *Site differentiates subtype B and C amino-acid consensus. †Known HLA-site association. Source data are provided as a Source Data file

**Table 5 Codons in *protease* showing evidence for HLA-associated selection**

| HLA | $\mathcal{C} = $ B | $\mathcal{C} = $ C |
|---|---|---|
| A*31 | | 57, 61, 62, 64 |
| A*66 | 35, 36*, 37 | |
| B*13 | | 62, 63*, 64 |
| B*44 | 35†, 37, 38, 39 | 35†, 36, 37, 38, 39, 41* |
| B*49 | 64, 65, 66, 67, 68 | 61, 62, 63*, 64, 65, 67, 68 |
| B*51 | | 37 |
| C*08 | 14 | |
| C*18 | 35 | 15*, 35 |

Sites defined by $\gamma_{h,i} > 2$ and lower 2.5% quantile > 1. Underline: Site of known drug resistance mutations. *Site differentiates subtype B and C amino-acid consensus. †Known HLA-site association. Source data are provided as a Source Data file

example, B*45 shows multiple signals of selection around codon 200 of *reverse transcriptase* (Fig. 5), yet there are no reported epitopes. This may reflect a historical bias towards studies being carried out in European-ancestry populations with subtype B viruses where some HLA alleles and viral epitopes are rare. However, we also note that results between the subtype B and subtype C consensus analyses are highly concordant, with a few interesting exceptions. For example, B*58 shows evidence for inducing strong selection away from the subtype B consensus around codon 122–123 of *reverse transcriptase* but not away from the subtype C consensus. Interestingly, this is a position of divergence between subtype B and subtype C viruses and B*58 is typically more common in populations with HIV-1 subtype C viruses, suggesting that B*58-induced selection pressure may have driven the fixation of the difference between subtypes. More generally, we found that average selection intensity away from the subtype C consensus in individuals carrying subtype B virus is significantly stronger than the average selection intensity away from the subtype B consensus (Supplementary Fig. 20: *protease* $p = 2.8 \times 10^{-5}$, *reverse transcriptase* $p = 2.0 \times 10^{-6}$; see Methods subsection: The impact of HLA on HIV-1 sequence evolution for details), though only *reverse transcriptase* was significant in the inverse setting ($p = 0.0058$, *protease* $p = 0.70$). Moreover, we estimate that the combined contribution of HLA-associated diversifying selection to non-

synonymous viral sequence change represents 59 and 77% of selection from subtype B and C consensus in *protease*, and 61 and 78% in *reverse transcriptase* (Methods subsection: Testing the impact of HLA on HIV-1 subtype differentiation, and Supplementary Table 8). In summary, we estimate that HLA-driven selection accounts for more than half of all HIV-1 coding changes and has contributed to diversification between subtypes.

To assess the relationship between sequence similarity between HLA alleles and the inferred HLA-induced selection profiles, we measured the concordance between dendrograms inferred from pairwise differences of classical HLA alleles protein sequences and dendrograms inferred from the estimated selection profiles (Supplementary Methods; Dendrograms of selection profiles and comparing topologies). We find that closely related HLA alleles have closely related selection profiles for HLA-A (odds ratio = 2.1; permutation p value = 0.017) and HLA-B (odds-ratio = 3.1; permutation p value = 0.014), but not HLA-C (odds ratio = 0.71; permutation p value = 0.71). However, deeper structure within the inferred trees shows little concordance. These findings provide biological validation for our results and suggest that future work to incorporate relationships between HLA allele would be valuable.

Finally, as noted recently[38], a number of sites both show evidence of HLA-driven selection as well as being associated with evolution in response to drug treatment (for example, around codon 70 in *reverse transcriptase* for B*45). Depending on whether the two types of selection act in the same or different directions, such pressures could either speed up or delay the origin of drug resistance. These results suggest that HLA genotype could be a predictor of the response to drug treatment.

## Discussion

Differences between hosts, such as in their immune system, or treatment received, can lead to host-specific selection regimes and subsequent adaptation by the pathogen. If within-host adaptation is at a cost to intrinsic fitness, then, on further transmission, selection may favour reversion to the fitter, ancestral state. To learn about such forces, the ideal experiment is to compare the genetic composition of the original infecting pathogen strain and that present sometime after successful infection across a large number of hosts that differ only in the factors of interest. While only strictly possible in experimental systems, observational approaches aim to learn about the same processes, but, to do so, have to make assumptions about the (unobserved) pool of pathogens to which infected individuals were exposed and the distribution of potential confounding factors. If assumptions are not met, power to detect true associations will be reduced and there is a risk of false positive results. The introduction of phylogenetic methods for association testing in pathogen genomics[17,20–22,25,26,39] provided a partial solution to many issues, by using genome-wide relatedness between pathogens as a proxy for correlation in unobserved confounding (such as population structure), similar to the use of principal components in the analysis of human genetic association[40]. Moreover, by developing explicit models of molecular evolution in response to host factors, it is possible to learn relative or even actual rates of escape and reversion[17]. However, phylogenetic methods have limitations. For example, they can be computationally expensive, meaning they are hard to apply to huge data sets; assumptions of homogeneity in rates over time and space typically are necessary; and, more fundamentally, recombination is widespread among pathogens.

To address these shortcomings, we have developed a model-based approach to association testing that exploits the availability of extremely large reference data sets on pathogen variation (but where there is no relevant metadata concerning the factors of interest), which are likely to contain genomes closely related to the samples of interest (with relevant metadata). Moreover, we argue that when identifying host-driven selection, most of the information lies near the tips of the trees as deeper comparisons have to integrate over many transmission events. By utilising the Li and Stephens[41] haplotype model, combined with previous work on modelling adaptive evolution and reversion[17,18,42] we have developed an approach that both dramatically increases power to detect associations and scales to huge data sets. Further, the framework can be extended in many ways, for example, to model the effects of covariates on recombination profile or intensity, or to incorporate hierarchical correlation structure between the selective effects of HLA types to boost power. Importantly, within the data analysed here, we estimate that between 7% (*protease*) and 17% (*reverse transcriptase*) of viral samples having undergone recombination at least once since their common ancestor with sequences in the reference data set.

The analysis of in vivo patterns of evolution provides a complementary approach to the in vitro study of drug resistance and immune evasion. The evolutionary response reflects the combination of fitness gains of escape and the fitness loss through modification. Hence, sites with strong resistance may have too strong a fitness cost to be typically selected for and weak resistance changes that have little fitness impact may be strongly selected for. Moreover, compensatory changes in the pathogen genome can potentially counteract fitness loss and hence appear as part of the resistance response. Our analysis of drug-associated evolution using data from the Los Alamos database identifies the majority of major DRMs; sites not identified typically result from failings of the model (variants that are more than one change away from the consensus codon, insertions and deletions). However, we also identify novel sites with compelling evidence for drug-associated selection, likely reflecting minor DRMs with small fitness loss and compensatory changes.

The analysis of HLA-associated selection presented here combines data from across six studies and nearly 3,000 HIV-1 sequences with linked patient HLA class I data. We show the method can recover well-known epitopes and associated escape mutations and that there is strong overlap between curated lists of epitopes and those identified here. However, we also found that simple in silico epitope prediction fails to provide a good predictor of the HIV-1 evolutionary response and multiple cases where curated epitopes do not appear to lead to a substantial selective pressure. We also showed a preponderance of signals of escape associated with HLA-B alleles, as opposed to HLA-A and HLA-C and identified a number of sites where differences between subtype consensus sequences collocate with sites selected by particular HLA alleles. Furthermore, by contrasting the aggregate HLA-associated selection in hosts of subtype B viruses with hosts of subtype C viruses, we observed differential signals of selection at sites distinguishing subtype B and C, supporting the hypothesis that differences in HLA allele frequency between populations could have contributed to the differentiation of HIV-1 subtypes[43]. As field-based sequencing of pathogens becomes feasible, so does the potential to accumulate vast data sets that can track spatial and temporal shifts in selection pressures affecting pathogens and their responses. Such studies will require new approaches to studying genetic association. While the method developed here is aimed at detecting within-host evolutionary responses, future extensions could include adaptation at a population level, for example through exploiting longitudinal sampling and/or spatial heterogeneity in treatment.

## Methods

**Motivation, justification and overview**. Our goal is to estimate the location and nature of host-associated selection upon the pathogen genome sequence through statistical analysis of association between host factor and pathogen genomic variation. We desire our method to have four properties: the ability to use all available pathogen sequence data, irrespective of whether host factors have been measured; the ability to measure the evidence for selection across the genome analysed in a hypothesis-free manner; the ability to combine information across neighbouring positions where appropriate; and the ability to account for confounding factors such as recombination and population structure. To meet these requirements we have developed a Bayesian model-based approach in which we use an approximation to the coalescent with recombination and codon-level selection[42]. In extension to the earlier work, we enable host-specific factors to influence patterns of variation and consider the case of two data sets. The first, $\mathbf{D}$, represents the collection of pathogen sequences for which host factor data (e.g., HLA genotype or drug treatment) is also available. The second, $\mathbf{D_B}$, represents a much larger data sets of pathogen sequences for which host factor data is not available. We use $\mathbf{D_B}$ to model the set of potential pathogen sequences that a host can be infected with (allowing for recombination) and assume that genetic differences between this reference panel and $\mathbf{D}$ largely reflects evolution within the host. Thus host-induced selection will result in an association between host factors and the evolutionary changes observed in $\mathbf{D}$. A cartoon of the underlying process and our model is shown in Fig. 1.

The justification of the approach is that if $|\mathbf{D_B}| \ll |\mathbf{D}|$ then members of $\mathbf{D}$ will typically coalesce very recently and approximately independently with members of $\mathbf{D_B}$. Thus the prior on $\mathbf{D}$ can be modelled with the Li and Stephens imperfect mosaic hidden Markov model (HMM), utilising a modified NY98 codon model[44]. To incorporate host-induced selection, we define a consensus codon $\mathcal{C}_i$ at site $i$ and model selection as a scaled increase in the rate of non-synonymous change away from $\mathcal{C}_i$. Conversely, we model reversion as a scaled increase in the codon substitution rate towards $\mathcal{C}_i$. To obtain emission probabilities we integrate over the distribution of coalescent time between a member of $\mathbf{D}$ and $\mathbf{D_B}$ assuming a coalescent model.

The HMM formulation enables efficient computation of the probability of observing a given sequence in $\mathbf{D}$ given current parameter values. The product over all members of $\mathbf{D}$ is used to approximate the joint probability $P(\mathbf{D}|\mathbf{D_B}, \mathcal{H}, \mathbf{\Theta})$, where $\mathcal{H}$ is host factor data for members of $\mathbf{D}$ and $\mathbf{\Theta}$ represents the parameters of recombination and codon substitution. To speed up computation, we *a priori* identify a subset of $\mathbf{D_B}$ for each member of $\mathbf{D}$ that is used to model its ancestor. The model parameters we aim to infer are: the synonymous transition rate, $\mu$, the $\frac{dN}{dS}$ ratio at each site, $\omega_i$, the recombination probabilities between neighbouring sites, $r_i$, the rate of reversion $\zeta_i$, and the host factor dependent scaling of escape rate at each

site, $\gamma_{h,i}$. We impose a piecewise constant prior on $\gamma_h$ to enable rapid exploration of $\gamma_{h,i}$ and regularisation of the inference. While motivated by the size distribution of epitopes, the piecewise constant prior is extremely flexible, for example encompassing a small number of discrete sites with high selection parameters. Other parameters (transition–transversion rate and fraction of codon substitutions more than one nucleotide change away) are estimated from external data. We use Markov chain Monte Carlo (MCMC) to sample from the posterior distribution for all parameters (See Supplementary Methods for details and MCMC moves). To guard against overfitting, we use shrinkage priors on the $\frac{dN}{dS}$ ratio and host factor associated selection coefficients. Similarly, the window structure on $\gamma_h$ borrows information across neighbouring sites, further guarding against overfitting in the absence of signal[42].

**Statistical and computational details**. In order to calculate the likelihood of the collection of parameters, $\Theta$, governing an evolutionary model, given a collection of sequence data $\mathbf{D}$, it is important to account for the non-independence of samples by considering the underlying genealogy $G$. For our purposes, we are not interested in estimating $G$. We, therefore, treat the genealogy as a nuisance parameter and integrate over it

$$P(\mathbf{D}|\Theta) = \int_G P(\mathbf{D}|\Theta, G)P(G)dG. \qquad (1)$$

Here, $P(G)$ is our prior on the genealogy $G$. $P(\mathbf{D}|\Theta, G)$ is the probability of the data under some parameter values $\Theta$ and genealogy $G$, which can be evaluated using Felsenstein's peeling algorithm[45]. Since the space of genealogies is very large, evaluating this integral is problematic. However, without recombination it is possible to estimate using a numerical summation approximation, as employed by BEAST[46] and MrBayes[47], for example. However, analysis is limited to hundreds or thousands of sequences depending on the properties of the analysed sequences. In the presence of recombination, $G$ becomes an ancestral recombination graph and so the state space of $G$ becomes far larger, making it impossible to use this approach in all but the simplest of models, and only for very small sequence data sets.

The solution that we use is to estimate the integral using an approximation to the coalescent with recombination[41]. The model we present has two components:

1. An approximation of the coalescent with recombination.
2. A model of codon substitution.

By combining these two components, we are able to approximate the probability of observing a given sequence in the presence of host factor dependent selection, and recombination. We make the assumption that each pathogen sequence is from a distinct individual. In practice, when applying the model to real data, we perform a number of data filtering steps to ensure this assumption is reasonable. We use the Li and Stephens imperfect mosaic HMM to approximate the coalescent with recombination. In the Supplementary Methods we describe the Li and Stephens model and the use of the forwards and backwards algorithms in evaluating likelihoods under this approximation.

In our inference problem we have two data sets. One query data set; $\mathbf{D}$ for which we have associated host factor information and another larger reference data set; $\mathbf{D_B}$ without host factor information. Recall from our model summary that we make two important assumptions:

1. All selection along the lineages connecting a member of $\mathbf{D}$ to $\mathbf{D_B}$ occurred within the host of the member of $\mathbf{D}$.
2. The reference data set $\mathbf{D_B}$ is a good approximation of the distribution of pathogens an individual can be infected with (which is reasonable when $|\mathbf{D_B}| \gg 0$).

These two assumptions allow us to apply the Li and Stephens approximation. In the Li and Stephen's HMM we require emission probabilities. For our model these will be probabilities of codon substitutions in the presence of some host factor. Making assumption 1 allows us to approximate $P(\mathbf{D}|\mathbf{D_B}, \Theta)$. This is because we have host factor information for the members of $\mathbf{D} = \{D_1, D_2, \ldots, D_n\}$ which we can use to determine the relevant emission probabilities. I.e., The first assumption allows us to use the sequence data in $\mathbf{D_B}$:

$$P(\mathbf{D}|\mathbf{D_B}, \Theta) = P(D_1|\mathbf{D_B}, \Theta)P(D_2|\mathbf{D_B}, D_1, \Theta) \\ \ldots P(D_n|\mathbf{D_B}, D_1, D_2 \ldots, D_{n-1}, \Theta). \qquad (2)$$

Using assumption 2, we may write

$$P(\mathbf{D}|\mathbf{D_B}, \Theta) \approx P(D_1|\mathbf{D_B}, \Theta)P(D_2|\mathbf{D_B}, \Theta) \ldots P(D_n|\mathbf{D_B}, \Theta) \qquad (3)$$

$$\approx \hat{\pi}(D_1|\mathbf{D_B}, \Theta)\hat{\pi}(D_2|\mathbf{D_B}, \Theta) \ldots \hat{\pi}(D_n|\mathbf{D_B}, \Theta). \qquad (4)$$

Here, $\hat{\pi}$ is the function describing the Li and Stephens approximation (described in the Supplementary Methods) with our model of recombination and codon evolution. In other words, assumption 2 results in the approximation that $\mathbf{D}$ is generated by independent realisations of recombination and mutation through $\mathbf{D_B}$. This avoids any need for averaging over orderings of $\mathbf{D}$, allowing rapid evaluation of the approximate likelihood.

Throughout the remainder of the Methods section we will refer to HLA-associated selection, but note that the method is general and extends to any host factor associated selection on pathogen sequence.

**Codon model of substitution**. We use a codon model of substitution model to describe sequence change. Importantly for our purposes, a codon model allows us to detect selection through non-synonymous nucleotide changes. At each site we are interested in detecting if these types of nucleotide changes away from some wild-type consensus sequence are enriched in a particular HLA background. We first write down the Nielsen and Yang (NY98) codon model (without HLA-associated selection)[44] and then extend it to incorporate the HLA types of the host transmitted to (a given member of $\mathbf{D}$). This will allow us to infer HLA-dependent selection along the pathogen sequence. Note that in Eqs. (5)–(24) we consider codon substitutions at a single site (and drop any site subscript). However, when we perform inference we will allow our codon model to be parameterised by distinct collections of parameters at each site.

In the NY98 codon model[44], the substitution rate from codon $i$ to codon $j$ is

$$q_{i,j} = \mu \begin{cases} \kappa & \text{if } i \text{ and } j \text{ differ by a synonymous transition} \\ 1 & \text{if } i \text{ and } j \text{ differ by a synonymous transversion} \\ \omega & \text{if } i \text{ and } j \text{ differ by a non-synonymous transversion} \\ \kappa\omega & \text{if } i \text{ and } j \text{ differ by a non-synonymous transition} \\ 0 & \text{otherwise,} \end{cases} \qquad (5)$$

where $\kappa$ is the relative rate of transitions to transversions and $\omega$ is the ratio of non-synonymous base changes to synonymous base changes assuming equal codon usage. The original model[44] weighted by the frequency of the codon switched to at that position to ensure reversibility. This is not a requirement for us. In fact, the process we want to model is not reversible, as we will include a boost in movement away from the consensus codon in the presence of a given HLA, and assume a distinct boost towards consensus independent of HLA (to model reversion). Finally $\mu$ is the scaled mutation rate parameter for the rate of synonymous transversions measured in units of $N$ generations (where $N$ is the effective population size). We note that while the original model was used to model fixed substitutions within species, here (as in many cases) the model is actually of how sequence diversity is generated within a population sample.

The exact probability of a substitution from codon $i$ to codon $j$ in time $t$ cannot be solved analytically, though we may estimate it by numerically evaluating the matrix exponential of the $64 \times 64$ instantaneous rate matrix described by Eq. (5) [42]. This is computationally expensive so we look for an approximation.

If the expected time until the first coalescence of a given lineage with all other sampled lineages $t$, is small (reasonable if the number of sampled sequences in $\mathbf{D_B}$ is very large), we can assume that the vast majority of codon changes occur through single-base changes (as the probability of observing more than one nucleotide change in a codon is low). This allows us to avoid matrix exponentiation and reduce computational cost. Consider a switch from codon $C_1$ to codon $C_2$. The total rate of substitution out of codon $C_1$ is

$$\left(\kappa\beta_S + \beta_V + \omega(\kappa\alpha_S + \alpha_V)\right)\mu =: \Lambda\mu, \qquad (6)$$

where $\beta_S$, $\beta_V$, $\alpha_S$ and $\alpha_V$ are the counts of the number of synonymous transitions, synonymous transversions, non-synonymous transitions, and non-synonymous transversions from $C_1$, respectively.

If the overall movement away from a particular codon $C_1$ is $\Lambda\mu$, then the probability a change has occurred in time $t$ is

$$P(\text{leave } C_1|\Theta, t) = \int_0^t \Lambda\mu \exp(-\Lambda\mu t)dt = 1 - \exp(-\Lambda\mu t). \qquad (7)$$

Here, $\Theta$ is the collection of parameters governing codon substitution. We can make an approximation by splitting up $P(C_1 \to C_2|\Theta, t)$. We consider two classes of moves from $C_1 \to C_2$: those which result from a single nucleotide change in time $t$ (e.g. AAA $\to$ AAG) and those that result from multiple nucleotide changes in time $t$ (e.g. AAA $\to$ ATA $\to$ AAA $\to$ AAG). For $C_1 \neq C_2$,

$$P(C_1 \to C_2|\Theta, t) = \sum_{i=1}^{\infty} P(C_1 \to C_2|i \text{ changes}, t)P(i \text{ changes}|\Theta, t) \qquad (8)$$

$$= P(\text{leave } C_1|\Theta, t) \sum_{i=1}^{\infty} P(C_1 \to C_2|\text{leave } C_1, i \text{ changes}, t) \\ \times P(i \text{ changes}|\Theta, t) \qquad (9)$$

$$\approx (1 - \exp(-\Lambda\mu t))(P(C_1 \to C_2|\text{leave } C_1, 1 \text{ change}, t)\phi \\ + P(C_1 \to C_2|\text{leave } C_1, >1 \text{ change}, t)(1-\phi)) \qquad (10)$$

$$\approx (1 - \exp(-\Lambda\mu t)) \\ \times \left(\phi P(C_1 \to C_2|\text{leave } C_1, 1 \text{ change}, t) + (1-\phi)\pi_{C_2}\right), \qquad (11)$$

where $\phi$ is the probability of undergoing a 1 step change from $C_1$ to $C_2$ and $\pi_{C_2}$ is the probability of observing codon $C_2$ at that position in the sequence.

Thus, for each codon $C_1$, we generate the following collection of probabilities for state changes at each site

1 step non-synonymous transition

$$P(C_1 \rightarrow C_2 | \mathbf{\Theta}, t) = (1 - \exp(-\Lambda \mu t)) \left( \phi \frac{\kappa \omega}{\Lambda} + (1 - \phi) \pi_{C_2} \right) \quad (12)$$

1 step synonymous transition

$$P(C_1 \rightarrow C_2 | \mathbf{\Theta}, t) = (1 - \exp(-\Lambda \mu t)) \left( \phi \frac{\kappa}{\Lambda} + (1 - \phi) \pi_{C_2} \right) \quad (13)$$

1 step non-synonymous transversion

$$P(C_1 \rightarrow C_2 | \mathbf{\Theta}, t) = (1 - \exp(-\Lambda \mu t)) \left( \phi \frac{\omega}{\Lambda} + (1 - \phi) \pi_{C_2} \right) \quad (14)$$

1 step synonymous transversion

$$P(C_1 \rightarrow C_2 | \mathbf{\Theta}, t) = (1 - \exp(-\Lambda \mu t)) \left( \phi \frac{1}{\Lambda} + (1 - \phi) \pi_{C_2} \right) \quad (15)$$

$\geq 2$ step change

$$P(C_1 \rightarrow C_2 | \Theta, t) = (1 - \exp(-\Lambda \mu t)) \left( (1 - \phi) \pi_{C_2} \right) \quad (16)$$

if $C_1 \neq C_2$, and

No change in codon at the site $1 - ((12) + \ldots + (16)) = (17)$

$$P(C_1 \rightarrow C_2 | \mathbf{\Theta}, t) = \exp(-\Lambda \mu t) + (1 - \exp(-\Lambda \mu t))(1 - \phi) \pi_{C_2}, \quad (17)$$

if $C_1 = C_2$. We estimate $\pi_C$ (for each codon $C$) and $\phi$ empirically using the Durban data set[48–50].

**Codon model of substitution: adding HLA-associated selection.** In order to account for HLA-associated selection we may choose to incorporate transmission and HLA proportions in the same way as the existing models[17,18]. Alternatively, we can consider a simpler model in which we are only concerned with the HLA type ($h$, say) of the sampled host in the smaller population. We explore the latter approach. For movement away from the consensus codon $\mathcal{C}$ at a given site, $\Lambda$ is modified to $\Lambda'$ in the presence of HLA type $h$

$$\Lambda' = \kappa \beta_S + \beta_V + \omega(\kappa \alpha_S + \alpha_V) \gamma_h. \quad (18)$$

So, as an example, for a non-synonymous transition from the consensus (which is predefined, in practice, we set it as either the consensus subtype B, or subtype C codon at the position) codon $\mathcal{C}$ to $C_2$, explicitly writing down the HLA dependence and using the approximation from Eqs. (13)–(17)

$$P(\mathcal{C} \rightarrow C_2 | h, t) = (1 - \exp(-\Lambda' \mu t)) \left( \phi \frac{\kappa \omega \gamma_h}{\Lambda'} + (1 - \phi) \pi_{\mathcal{C}} \right), \quad (19)$$

$\gamma_h$ is a scaling of the non-synonymous/synonymous substitution ratio $\omega$ associated with HLA type $h$, away from $\mathcal{C}$. This weighting provides extra selection away from the consensus codon $\mathcal{C}$ if $\gamma_h > 1$ and less if $\gamma_h < 1$ to create a new total rate of movement out of the consensus codon $\mathcal{C}$ in the presence of HLA type $h$, $\Lambda'$. Thus an association between a particular HLA type and non-synonymous change from consensus at the site would result in $\gamma_h > 1$. Notice that no selection is applied to synonymous codon changes.

Extending to an HLA profile of a host $\mathbf{H} = \{h_1, h_2, \ldots, h_n\}$

$$P(\mathcal{C} \rightarrow C_2 | \mathbf{H}, \mathbf{\Theta}, t) = (1 - \exp(-\Lambda' \mu t)) \left( \phi \frac{\kappa \omega \prod_{h \in \mathbf{H}} \gamma_h}{\Lambda'} + (1 - \phi) \pi_{\mathcal{C}} \right), \quad (20)$$

We also wish to incorporate reversion into our codon model. Given the state of the codon under consideration, there is some set of nucleotide substitutions resulting in $\mathcal{C}$ at that position. It is these substitutions which we wish to scale through reversion parameters. We let $\zeta$ denote the scaling of selection due to reversion at a particular site. We assume that each $\zeta$ is independent of the host's HLA profile, $\mathbf{H}$. By doing so, we are assuming that in the absence of HLA pressure there is a selection towards the consensus codon $\mathcal{C}$ at each site. Note that reversion only acts on non-synonymous changes that result in the consensus codon $\mathcal{C}$. Consequently, the parameterisation is identifiable and we are able to distinguish $\omega_i$ and $\zeta_i$ in the product $\omega_i \zeta_i$.

Consider the case in which a non-synonymous transition results in the consensus codon, $\mathcal{C}$, from some codon $C_1$. Then $\Lambda$ is modified to $\Lambda'$:

$$\Lambda' = \kappa \beta_S + \beta_V + \omega(\kappa(\zeta + (\alpha_S - 1)) + \alpha_V). \quad (21)$$

Reversion is therefore modelled by

$$P(C_1 \rightarrow \mathcal{C} | \mathbf{H}, \mathbf{\Theta}, t) = (1 - \exp(-\Lambda' \mu t)) \left( \phi \frac{\kappa \omega \zeta}{\Lambda'} + (1 - \phi) \pi_{\mathcal{C}} \right). \quad (22)$$

$\zeta$ is a scaling of the non-synonymous/synonymous substitution ratio $\omega$ towards $\mathcal{C}$. This weighting provides extra selection towards the consensus codon $\mathcal{C}$ if $\zeta > 1$ and less if $\zeta < 1$. We assume that the same $\zeta$ scales all the potential 1 step non-synonymous base changes to $\mathcal{C}$.

We have now created a codon model which accounts for reversion and HLA-associated selection within a host, and can now determine emission probabilities; $\varepsilon_{j,i}$. We define $\varepsilon_{j,i}$ as the probability of copying the codon at position $i$ from

sequence $j$ (see Supplementary Methods for background details). Therefore, by evaluating

$$P(C_1 \rightarrow C_2 | \mathbf{H}, \mathbf{\Theta}) = \int_0^\infty P(C_1 \rightarrow C_2 | \mathbf{H}, \mathbf{\Theta}, t) P(t) dt, \quad (23)$$

which is the probability that a change from codon $C_1$ to $C_2$ has occurred in the time to coalescence with the reference set, we can substitute in the relevant codon change ($C_1 \rightarrow C_2$) and determine $\varepsilon_{j,i}$. We integrate out $t$ assuming the coalescent with fixed population size. In the standard coalescent with a fixed population size, the distribution of the divergence time between a sequence and its closest relative from among $k$ other sequences may be approximated by an exponential distribution with rate $\frac{k}{2}$ $\left(P(t) = \frac{k}{2} \exp\left(-\frac{k}{2} t\right)\right)$[42] (see Supplementary Notes for a derivation of the mean).

This is simply a case of substituting in and integrating. For example, for a non-synonymous transition from the consensus,

$$P(\mathcal{C} \rightarrow C_2 | \mathbf{H}, \mathbf{\Theta}) = \left( \phi \frac{\kappa \omega \prod_{h \in \mathbf{H}} \gamma_h}{\Lambda'} + (1 - \phi) \pi_{C_2} \right) \left( 1 - \frac{k}{k + \Lambda' \mu} \right). \quad (24)$$

Notice that in evaluating this integral we are focused on sequence evolution between a member of $\mathbf{D}$ and its nearest neighbour in $\mathbf{D}_B$. Only considering nearest neighbours in the genealogy has two clear advantages:

1. We only use the parts of the genealogy that contain the most information about HLA-dependent selection within a sampled host: terminal branches.
2. The resulting reduction in computation time allows us to use far more sequence data. Using more sequence data will shorten terminal branches and so increase our power to detect HLA-associated selection.

Using very large data sets is important for our method. As terminal branches shorten through increased sampling, assumption 1 (that selection occurs in the member of $\mathbf{D}$ along the lineage connecting each member of $\mathbf{D}$ to its nearest neighbour in $\mathbf{D}_B$) becomes more reasonable as we may assume that no or very few transmissions occur along the lineage connecting sequences in $\mathbf{D}$ to their nearest neighbour in $\mathbf{D}_B$.

In Eqs. (18)–(24) we considered a single host with an HLA type $h$ or collection of HLA types $\mathbf{H}$. We can trivially extend this to obtain the desired approximation of the likelihood given in Eq. (3). Let $\mathcal{H}$ be the collection of HLA profiles of the associated hosts for sequences in $\mathbf{D}$. Let $\mathbf{H}_j$ denote the HLA profile of host $j$. Then

$$P(\mathbf{D} | \mathbf{D}_B, \mathcal{H}, \mathbf{\Theta}) \approx \prod_j \hat{\pi}(D_j | \mathbf{D}_B, \mathbf{H}_j, \mathbf{\Theta}). \quad (25)$$

We have now created a model which accounts for recombination using the Li and Stephens[41] approximation and incorporates both reversion and HLA-associated selection. We perform inference using MCMC.

**Restricting $\mathbf{D}_B$: sample-specific reference data sets $\mathbf{D}_{B_j}$.** In order to determine the posterior probability of a new state within the MCMC quickly, we keep track of two large arrays for each query sequence (as well as other quantities required to evaluate the likelihood). This requires storing two $|\text{codon sequence}| \times |\mathbf{D}| \times |\mathbf{D}_B|$ arrays of doubles. When evaluating the likelihood of observing ~2,000 query sequences from a collection of ~60,000 reference *protease* sequences from the Stanford drug resistance database, over 300 GB of RAM was required! We require a method to reduce the memory required to perform inference.

We restrict $\mathbf{D}_B$ to a different subset $\mathbf{D}_{B_j}$ for each $D_j \in D$. We make the approximation

$$P(\mathbf{D} | \mathbf{D}_B, \mathcal{H}, \mathbf{\Theta}) \approx \prod_j \hat{\pi}(D_j | \mathbf{D}_B, \mathbf{H}_j, \Theta) \approx \prod_j \hat{\pi}(D_j | \mathbf{D}_{B_j}, \mathbf{H}_j, \mathbf{\Theta}). \quad (26)$$

If the sequences we choose for each $\mathbf{D}_{B_j}$ are similar to the true ancestors of each of the $D_j$, then this will be a good approximation.

The simple approach we use to restrict $\mathbf{D}_B$ is by considering the closest $n$ sequences to $D_j$ by Hamming distance (the Hamming distance between two strings is the total number of differences between the strings, giving each difference equal weighting). A drawback of using Hamming distance to restrict $\mathbf{D}_B$ is that in the presence of high recombination rates, one can imagine sequences which should be included in $\mathbf{D}_{B_j}$ but are excluded on the basis this metric.

For the small amount of overhead, this sample-specific restriction reduce the computational time and memory footprint required by the programme to the extent that a run of 1,500,000 updates analysing ~1,000 query sequences over a codon sequences of length ~100, considering the closest 100 reference sequences according to some metric may be performed in ~24 h on an Intel i7 desktop machine using <1 GB of RAM.

**MCMC inference regime.** The structure of the model we have presented lends itself to inference using MCMC to sample from the posterior. For our MCMC implementation, we use the Metropolis Hastings algorithm. We fix $\kappa$, $\phi$ and $\pi_C$ for each codon $C$ at empirical estimates. We perform MCMC moves on the following collection of parameters:

- The recombination probability $r_i$ between neighbouring sites $i$ and $i + 1$.
- The non-synonymous/synonymous substitution ratio in the absence of

reversion or HLA-associated selection at each site, $\omega_j$.
- Scaling of selection due to reversion at each site, $\zeta_i$.
- HLA-associated scalings of selection at each site, $\gamma_{h,i}$.

In our sampling scheme we allow $\zeta_i$ and $\gamma_{h,i}$ to vary via piecewise constant functions across the codon sequence, which we call selection windows. This is analogous to the 'block-like' model used by Wilson and McVean[42]. Imposing this window structure allows information about HLA-dependent selection (and reversion) to be combined across sites. This makes sense in a biological setting where sites close together in the coding sequence may result in amino acids lying in the same epitope or close together in the protein structure. The window model will act to smooth across sites, but will be overwhelmed if a given site is subject to strong HLA-dependent selection[42]. How strict the window model is depends on the expected number of windows, which is controlled by the parameter $p_w$ (see the merge and split moves in Supplementary Methods MCMC moves subsection, and Wilson and McVean[42]). If $p_w = 1$, the number of selection windows is equal to the number of sites, if $p_w = 0$ then there is just one selection window. The piecewise constant prior is extremely flexible. Using a piecewise constant prior in combination with MCMC moves to shrink/expand and merge/split these blocks allows us to explore the space of HLA-associated selection rapidly while maintaining a far smaller number of parameters than considering HLA-associated selection site by site. See Supplementary Methods for MCMC move details.

**Simulation study 2: simulating a sampled birth–death process.** For a second large simulation study, we consider a more realistic generative process for creating our reference and query data sets. Rather than using an existing large reference data set and simulating query sequences under our model, we generate an instance of a sampled birth–death process. We then simulate sequence data down the resultant sampled birth–death tree.

We simulate the sampled tree backwards in time together with unseen transmission events using the following procedure:

Let birth rate be $\lambda$ and the death rate be $\delta$. Thus, backwards in time $\lambda$ is a death rate, and $\delta$ a birth rate. Let $N$ and $M$ denote the total number of infected individuals and sampled infected individuals respectively. Setting $r$ as the sampling proportion for extinct lineages, we consider a collection of competing Poisson processes.

Set $t = 0$, and let the total number of infected individuals and sampled infected individuals at the present be $N = \bar{N}$ and $M = \bar{M}$, respectively.

While $M > 0$, sample the time of the next event $t \to t + \bar{t}$ where $\bar{t} \sim Exp(N(\lambda + \delta))$. First, we determine whether this event is a birth or a death. Sample $u \sim U[0,1]$.

- Birth. If $u < \frac{\lambda}{\lambda + \mu}$, then $N \to N - 1$ and there are three possibilities here with the following probabilities.

  An unseen lineage coalesces with unseen lineage with probability $\left(1 - \frac{M}{N}\right)^2$.
  An unseen lineage coalesces with a seen lineage with probability $\left(1 - \frac{M}{N}\right)\frac{M}{N}$, and we record the time $t$.
  A seen lineage coalesces with a seen lineage with probability $\left(\frac{M}{N}\right)^2$, $M \to M - 1$, and we record the time $t$.

- Death. Else $\left(u \geq \frac{\lambda}{\lambda + \mu}\right)$, and $N \to N + 1$.

  If $\hat{u} \sim U[0,1] < q$ then the death event is sampled $M \to M + 1$, and we store the time of the event $t$.

Given that birth and death events are lineage independent, we may just use the time information to then generate the a tree. Code is available at https://github.com/astheeggeggs/mcqueen. We note that this generative process is based on the equations laid out in the Supplementary Materials of Palmer et al.[17] and Frost and Volz[51], and that it is similar to the backwards episodic birth–death process algorithm outlined by Stadler[52], except that we have a fixed birth and death rate over the entirety of the tree, store unseen transmissions, and do not have mass extinction events.

We now have a sampled birth–death tree. Given our knowledge of all the transmission events along the lineages of this sampled birth–death tree, we may assign HLA types to each node by passing the information down the tree from the root. Note that at each coalescence event, one of the daughter lineages will be in the same host, whilst the other will represent the transmission to a new host and HLA environment. All unseen transmission events which we count are transmissions to a new host, so also coincide with a change in HLA environment.

We then simulate sequence change down this tree conditional on all the HLA information, using the codon substitution model outlined in the methods subsection: Codon model of substitution, again by passing a sequence at the root and tracking changes down the tree. Finally, we sample $m$ of these sequences at the leaves to be our query sequence set with associated host HLA information, and the remainder to define our reference sequence data set, after throwing away the host HLA information. In this simulation study, we set $N = 1{,}000{,}000$, $M = 100{,}000$, $q = 0.1$, and $m = 1{,}460$, and use the same selection profiles, but increase the

number of flat selection profiles to 16, so that the total frequency of a given HLA allele more closely resembles that seen in the data, which is important for this tree based simulation of HLA-associated evolution.

**Simulation study 3: the effect of population differentiation.** Differentiation between the query and reference data sets can potential lead to reduced performance of estimators due to two factors: confounding between covariates that are non causal (for example differences in both HLA allele frequency and viral genetic diversity due to genetic drift) and relatedness within the query population leading to poorly calibrated estimates. To assess the impact of population differentiation on performance of our estimator we carried out a parametric bootstrap simulation, using viral sequence data from the Botswana study ($n = 343$, *protease* only) to simulate ancestors infecting a set of hosts with HLA allele frequencies drawn from the Botswana population. We used the evolutionary parameters estimated within this study (including the HLA-associated selection profiles, $\frac{dN}{dS}$, and recombination probabilities) and performed 100 independent simulations of 1,500 query sequences with associated HLA information. For each simulation we then performed inference using a range of reference data sets and also considered a leave-one-out (LOO) strategy in which the reference data are augmented by all sequences from the query data set except for the sequence under consideration. These will typically be the most closely related to the query data. The reference data sets we consider are:

a. $\mathbf{D_B}$ = The sequences actually simulated from, under the model (Botswana sequence data). This represents a gold-standard reference data set.
b. $\mathbf{D_B}$ = All sequence data available in public databases for which no HLA information is available ($n = 162{,}901$).
c. $\mathbf{D_B}$ = The simulated sequences, using a LOO strategy for each query.
d. $\mathbf{D_B}$ = $a) + b)$, using a LOO strategy.

We display plots of inference results for a common and rare HLA type for each HLA class I molecule (A–C) in Supplementary Figs. 9–11.

To compare the impact of relatedness on our ability to perform inference, we compare the mean RMSE of the median HLA-associated selection estimates across sites, weighted by the frequency of the HLA types in our simulated data sets. Results are displayed in Supplementary Table 9.

As expected, the gold standard $\mathbf{D_B}$, $a)$ is the most accurate. We find that the reference data set that we have aggregated, $b)$ performs almost as well, with the addition of the simulated sequences to the reference data set performing worse. This is likely due to the approximate nature of the model used to simulate the sequence data, but nevertheless is encouraging to see that our current approach recapitulates parameters almost as well as if we had access to the best possible query data set, $a)$.

We also examined the effect of restricting the reference data set to the subset Los Alamos reference data (where we had sampling country information). We then checked to see if removal of all Botswana samples ($n = 73$) from this Los Alamos reference set dramatically reduced our accuracy. We saw similar results for inference using these two reference sets.

These results indicate that our method is robust to population differentiation and relatedness within the query data set. Importantly, the effect of removing all reference data from Botswana has only a marginal impact on accuracy. The LOO strategy does not typically improve performance, though clearly enables use of the method when no (or only very distantly related) reference data is available. Users of our software (available at https://github.com/astheeggeggs/mcqueen) can set $\mathbf{D_B}$ to be created using a LOO approach by using the -q or –separate_reference_fasta flags.

**Testing overlap with A-list and B-list epitopes.** We permute the labels of HLA types for a class I gene 1,000,000 times, and count the number of overlaps in the putative epitope sets for each permutation. This leads to an estimate of an alternative null distribution (by shuffling we account for possibility that parts of the region may be epitope rich across HLA types). We then compare this distribution of counts of overlapping sites to the observed number of epitope overlaps with our putative selected sites to obtain $p$ values, by determining the proportion of the shuffled sets with at least as many overlaps as the real data.

**The impact of HLA on HIV-1 sequence evolution.** To investigate the contribution of diversifying selection due to HLA pressure, we first evaluate the average HLA-associated selection away from consensus at each site $i$, for each individual $j$ in the sample using the median estimates from the MCMC analysis

$$A_{j,i} = \omega_i \left( \kappa \alpha_{S,i} + \alpha_{V,i} \right) \prod_{h \in \mathbf{H}_j} \gamma_{h,i}, \tag{27}$$

where $\alpha_{S,i}$ and $\alpha_{V,i}$ are the number of non-synonymous transitions and non-synonymous transversions from the consensus codon at site $i$, respectively.

We then determine the collection of $A_{j,i}$ for which there is increased selection away from consensus due to HLA-associated selection: $\left( \prod_{h \in \mathbf{H}_j} \gamma_{h,i} \right) > 1$. We then evaluate the fraction of the total mutation rate away from consensus that this

represented

$$\frac{\sum_{i,j:\left(\prod_{h\in\mathbf{H}_j}\gamma_{h,i}\right)>1}A_{j,i}}{\sum_{i,j}\left(\omega_i\left(\kappa\alpha_{S,i}+\alpha_{V,i}\right)\prod_{h\in\mathbf{H}_j}\gamma_{h,i}+\left(\kappa\beta_{S,i}+\beta_{V,i}\right)\right)},\tag{28}$$

and the fraction of the mutation rate that resulted in an amino-acid substitution

$$\frac{\sum_{i,j:\left(\prod_{h\in\mathbf{H}_j}\gamma_{h,i}\right)>1}A_{j,i}}{\sum_{i,j}\left(\omega_i\left(\kappa\alpha_{S,i}+\alpha_{V,i}\right)\prod_{h\in\mathbf{H}_j}\gamma_{h,i}\right)}.\tag{29}$$

We find that this represents 18 and 28% of all mutations (59% and 77% of non-synonymous substitutions) away from subtype B and C consensus in *protease*, respectively. In *reverse transcriptase* this represented 18% and 32% of all mutations (61% and 78% of non-synonymous substitutions) away from subtype B and C consensus, respectively.

**Testing the impact of HLA on HIV-1 subtype differentiation**. We wish to examine whether amino-acid differences between HIV subtype B and C may have been driven by differences in HLA frequency distributions. To do this, we first use RIP[53] to determine, for each query sample, the viral subtype within each of the genomic regions analysed. At each site we then calculate (using the inferred evolutionary parameters) the average selection away from each subtype consensus in those hosts harbouring the subtype B and C viruses separately. In this way, we obtain an approximation of the average HLA-associated selective pressure away from the subtype B/C consensus in the geographic regions in which subtype B and subtype C viral sequences predominantly reside. For example, letting $S_C$ be the set of subtype C viral samples, $\mathbf{H}_j$ be the HLA profile of the host in which viral sequence $j$ resides, and $\gamma^B_{h,i}$ denote HLA $h$ associated selection away from subtype B at site $i$

$$A^B_{i,C}:=\frac{1}{|S_C|}\sum_{j\in S_C}\left(\omega_i\left(\kappa\alpha_{S,i}+\alpha_{V,i}\right)\prod_{h\in\mathbf{H}_j}\gamma^B_{h,i}\right)\tag{30}$$

is the selective effect away from subtype B consensus at site $i$ of HLA alleles averaged over regions in which subtype C predominates. Results are shown in Supplementary Fig. 20.

To determine if there is an elevated HLA-associated selective effect on sites that differ between subtypes B and C, we performed permutation tests. For example, consider selection away from subtype B consensus. We evaluate

$$\sum_{i\in D}\left(A^B_{i,B}-A^B_{i,C}\right),\tag{31}$$

where $D$ is the set of sites at which subtype B and subtype C differ at the amino-acid level. We then randomly shuffled site labels and re-evaluated this quantity 1,000,000 times to obtain a $p$ value. Resultant $p$ values for protease and reverse transcriptase are shown in Supplementary Table 8.

**Reporting summary**. Further information on research design is available in the Nature Research Reporting Summary linked to this article.

## Data availability
Host HLA data used in this study are available from the corresponding author upon reasonable request. All reference data sets; $\mathbf{D}_B$, and query data sets for drug-associated selection inference are available in the mcqueen paper repository at www.github.com/astheeggeggs/mcqueen_paper. Source data underlying Figs. 2–5, Supplementary Figs. 2–20, Tables 3–5 and Supplementary Tables 4–6 and 9 are available in the source data file. Raw MCMC output for simulation studies, HLA, and drug inference are available at www.github.com/astheeggeggs/mcqueen_paper.

## Code availability
Code implementing our methods is available at www.github.com/astheeggeggs/mcqueen with no restriction to access, under the MIT licence.

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

## Acknowledgements

Acknowledgements Funded by an EPSRC studentship to D.P., the Li Ka Shing Foundation, Wellcome Trust grant 100956/Z/13/Z to G.M., Wellcome Trust grant WT104748MA to PJRG, National Institutes of Health grant RO1AI46995 to PJRG, National Institute of Allergy and Infectious Diseases grant U01-AI066454 to RS, European Union grant SANTE/2007/147-790 to D.G. and C.V. J.F. funded by the MRC. Computation used the Oxford Biomedical Research Computing (BMRC) facility, a joint development between the Wellcome Centre for Human Genetics and the Big Data Institute supported by Health Data Research UK and the NIHR Oxford Biomedical Research Centre. The views expressed are those of the author(s) and not necessarily those of the NHS, the NIHR or the Department of Health. We thank Hannah Roberts and Jessica McGillen for thoughtful discussions.

## Author contributions

G.M. and D.S.P. conceived the model, analysed and interpreted the results. S.F., J.F., D.G., P.G., K.H.G.H., A.O., R.P., R.S. and C.V.V. generated and shared the viral sequence data and host HLA information. D.S.P. and I.T. implemented the model. A.R.M. aided in the interpretation of results. G.M. and A.R.M. supervised the research. D.S.P. and G.M. wrote the paper.

## Additional information

**Competing interests:** The authors declare no competing interests.

