## [Peer Review File · Nature Communications]

Reviewers' comments:

Reviewer #1 (Remarks to the Author):

I like the model in this manuscript. It is very much in the Oxford school – building on previous work of the last author and others, but is timely, intricate, is described clearly and appears from the description to be well implemented. The improvements in performance over alternative approaches for simulated data appear impressive, even allowing for some natural bias towards simulation scenarios that are likely to favour the author's approach.

As someone who does not know the HIV literature, and therefore may not appreciate the extent of the advance made, I was less sure what to make of the analysis of real data. My prior is that selection for drug resistance is likely to be a relatively easy problem - and it is not clear to me as an outsider how independent the previously defined drug resistance mutations are to the data analysed in the paper. The concordance is fairly high but a randomization test is not a very stringent criterion, especially since mutations that are implicated in drug resistance presumably are not random subset of all mutations (e.g. excluding those that are observed very rarely).

Evolution in response to HLA alleles is a harder problem and the results seem to be very noisy, contrasting greatly for example with the simulations in Figure 2 which implies that many biological parameters can be estimated with high precision. Figure 5 is complicated and hard to understand and mainly seems to indicate how difficult the problem is and how unreliable most or possibly all approaches probably are. Why do figures 2 and 5 give such different impressions, I wonder!? Maybe the estimate the authors make are the only good ones, in which case it is a tremendous advance but I'd like to have some strong arguments to make me believe this. Even if the results are correct, if it is not possible to validate them, what use are they?

Although I only have a limited number of concrete suggestions and do not think the general criticism above is necessarily fair, I would encourage the authors to think more creatively about what further information can be harvested from the model – perhaps biological parameters that are entirely inaccessible using other approaches and what additional analyses can be done to confirm that the inferences are biologically meaningful. In any case, tightening up the logic of the real data analysis should be helpful to readers like me.

A first major aspect that would profit from additional attention is estimation of how related the samples in the database DB are to those in the phenotyped set D. My intuition is that the degree of relatedness is critical to the power of the method and indeed information from strains in D that are closely related to those in DB is much larger than from those with even slightly more distance relations. Specifically, the signal of adaptation that is being searched for is the changes in the most recent infection and therefore the specificity of the signal decreases with each extra transmission between D and DB. This is relevant especially because the degree of genetic relatedness can be estimated from the HMM. It seems to me that the information from strains at different distances to DB could be weighted differently in inference. This factor can also affect the power of inferences and explain why some phenotypes are easier to study than others etc; different phenotype sets can very easily differ systematically in their relatedness to the same database. All of this can be investigated empirically and might possibly explain some of the difference between the real and simulated data? It would also be useful to know for experimental design purposes which strains contributed substantially to the selection signatures. It might even be useful in validation.

I would also like to know what proportion of substitutions inferred by the HMM are likely, under the model, to be explained by selection linked to the measured phenotypes. This might be a useful statistic to output to get an impression of what is going on. Other indication of measured effect sizes would also/alternatively be welcome.

A second major aspect is confounding by population structure. The introduction dwells on this as being an important factor but while the method does seem to provide some protection from the most severe sources of confounding, I do not think it eliminates the risk entirely. However, this is not discussed at

all later in the paper. Especially if the focus of the manuscript is new methodology, I think there should be detailed discussion of the issue and also a demonstration of what empirical evidence can be brought to bear to investigate it. Specifically, it seems relevant how the members of D are related to each other and how this compares with their relationships with the strains they are inferred to get ancestry from in DB.

A third more speculative aspect is that it would be interesting to find out more about whether recombination could be related to other evolutionary processes or phenotypes.

Reviewer #2 (Remarks to the Author):

Palmer and colleagues describe a joint statistical model for simultaneously estimating parameterized escape (e.g. from HLA or antiretrovirals), reversion, and selection from unknown sources while correcting for population structure and recombination. The key observation is that the existence of massive reference datasets allow for some simplifying assumptions that make such estimation more tractable than possible with smaller datasets. The model described is elegant, the experiments generally well done, and the manuscript uncommonly well written. While I do have some comments on how the manuscript should be improved, overall it is an important and useful contribution.

The manuscript is logically broken into three sections, and I will comment on them as such.

1) The model. Overall, the model is well described and mostly well motivated. I do have some modest concerns however.

a. I am unconvinced of the motivation for the block prior. In the case of ARV, is there evidence that we should expect selection from a specific drug to uniformly impact a window of a protein? In the case of HLA, an epitope provides such a window, but most of the available evidence appears to suggest that actual selection occurs at a small number of discrete sites within the epitope. The implementation of the window is a bit confusing to follow, and I expect computationally expensive while possibly increasing the variance on parameter estimates. Is it justified?

b. The sheer number of parameters estimated is large and I worry about overfitting. While the datasets are very large, when considering, e.g., HLA alleles that occur in 10 individuals (their lower limit in the real data), the effective size for parameter estimation is very small. Is this a concern?

c. The authors had seemingly small comments about truncating their data that were not discussed. E.g., using 2 digit HLA resolution (when available evidence suggest a majority of escape is specific to 4 digit alleles), or conversely not grouping classes of drugs. Would their model be amenable to a hierarchical prior, which has the potential to automatically select the level of grouping most appropriate for a given codon? A hierarchical prior could also provide shrinkage to deal with overfitting.

d. The separation of D from Db was not well motivated. One could easily construct cases where the majority of samples from a given region are in the query set, not reference. E.g., in their HLA data, I imagine the majority of available samples from the city of Durban are in their query set, which would violate their assumption that the reference set is an effective estimate of the population of available sequences at time of transmission. But could not D be included in Db (with the constraint that a specific sequence i cannot be derived from itself)?

e. The parameterization of reversion appears to be underdetermined. If I'm following the model

correctly, reversion is “independent of HLA” (shouldn’t it be only among individuals without a given HLA?), then multiplied into ω_i (S36). But ω and ζ are estimated independently for each site. How is ω_i and ζ_i then identified? Does this explain why reversion is not well estimated in the synthetic data?

f. Please double check S19. Transition and transversion should be swapped for the last two.

2) Synthetic experiments

a. The authors should explore what happens when their assumptions are not met, as will undoubtedly be the case to some degree. E.g., what happens if data are generated from the Botswana sequences, then all Botswana sequences are removed from Db (simulating the case where the query sequences make up the majority of known sequences for a region)? How massive does Db need to be? They ran their Birth-Death model sampling 10% = 100k sequences. What if they had 10k? 1k? The question is, is HIV Pr/RT and Influenza H/N the only datasets we can use this for, or will it work for other viruses (or proteins) with less data? When there is not enough Db data, or when the data in Db is not similar enough to D, does the model fail gracefully?

b. Were the tests of “common” vs “rare” alleles independent, and containing a single HLA? Is this overly optimistic compared to including a realistic number of Hla alleles? It seems it may be easier for the model to converge if the number of alleles is small.

c. In terms of comparisons, I would expect that the present model will benefit most compared to other models when the query size |D| is small, as this model can leverage Db? E.g., in the Real HLA set considered, 3k sequences in D is enough that I imagine most of the models will have quite a bit of statistical power.

3) Real data

a. What were inferred recombination rates? Is there evidence this is an important feature of the model?

b. I do not follow the motivation for adding 2k sequences from Db to D for the drug association study.

c. Why are HLA A-, B- and C- treated independently? There was nothing in the model description that suggests to me these can’t be jointly estimated, unless it’s a matter of reducing the number of parameters? (in which case, a hierarchical prior might help)

d. The authors should probably not claim this to be “the most complete and comprehensive to date...covering nearly 3,000” HLA-linked Pr and RT sequences. E.g., Carlson (and some of the present authors) 2016 (PMID 27183217) included over 3k HLA-linked sequences from Gag, Pol and Nef. Comparing the results might be an interesting exercise, especially considering the subtype C sequences for the two studies largely overlap.

e. In the discussion, the authors note that some HLA escapes are at sites of subtype consensus differences. This observation has previously been made for Gag, where it was further noted that HLAs common to subtype C (e.g.) were selecting for the C consensus (Carlson et al, 2008; PMID 19023406). Do the authors similarly find sites where alleles are associated with selection of consensus, which I think would manifest as $\gamma \ll 1$?

f. It might be noted that including the early infection SPARTAC cohort may be reducing their power, as

these individuals may not have been infected long enough for escape to have occurred, thereby diluting the escape signal (e.g. Martin et al, 2014, PMID 25212686).

Response to Reviewers

Substantial changes in and additions to the text are indicated by blue text within the manuscript. We indicate below where changes have been made and, for some responses, paste the altered or additional text in grey.

Reviewer 1

I like the model in this manuscript. It is very much in the Oxford school – building on previous work of the last author and others, but is timely, intricate, is described clearly and appears from the description to be well implemented. The improvements in performance over alternative approaches for simulated data appear impressive, even allowing for some natural bias towards simulation scenarios that are likely to favour the author's approach.

We thank the reviewer for their positive comments.

As someone who does not know the HIV literature, and therefore may not appreciate the extent of the advance made, I was less sure what to make of the analysis of real data. My prior is that selection for drug resistance is likely to be a relatively easy problem - and it is not clear to me as an outsider how independent the previously defined drug resistance mutations are to the data analysed in the paper. The concordance is fairly high but a randomization test is not a very stringent criterion, especially since mutations that are implicated in drug resistance presumably are not random subset of all mutations (e.g. excluding those that are observed very rarely).

The aim of including the section on drug resistance mutations (DRMs) was to validate the method within a real-world setting. That is, we were evaluating the method on its ability to recover known features of DRMs, rather than detect novel associations. Importantly, within the set of sites labelled as drug resistance causing within the Stanford Drug Resistance Database (hivdb.stanford.edu), many (referred to as 'Major DRMs') have been validated through in vitro resistance assays. Hence although there is some overlap between the data used here and previous analyses of drug resistance (hence candidates for DRMs), only those with independent experimental validation considered in the enrichment study. In addition to reporting enrichment analyses we have also explored all apparent false positives and false negatives (Supplementary Tables 2 and 3) and provide a summary of the results in the text. Our detailed evaluation revealed the complexities of assigning DRM status to specific changes, as well as highlight a small number of modelling issues.

Evolution in response to HLA alleles is a harder problem and the results seem to be very noisy, contrasting greatly for example with the simulations in Figure 2 which implies that many biological parameters can be estimated with high precision. Figure 5 is complicated and hard to understand and mainly seems to indicate how difficult the problem is and how unreliable most or possibly all approaches probably are. Why do figures 2 and 5 give such different impressions, I wonder!?! Maybe the estimate the authors make are the only good ones, in which case it is a tremendous advance but I'd like to have some strong arguments to make me believe this. Even if the results are correct, if it is not possible to validate them, what use are they?

Although I only have a limited number of concrete suggestions and do not think the general criticism above is necessarily fair, I would encourage the authors to think more creatively about what further information can be harvested from the model – perhaps biological parameters that are entirely inaccessible using other approaches and what additional analyses can be done to confirm that the inferences are biologically meaningful. In any case, tightening up the logic of the real data analysis should be helpful to readers like me.

Given the number of different factors affecting within-host evolution and the highly stochastic nature of the processes involved, inference is likely to be noisy. Figure 2 summarises a series of independent simulations and shows how – on average – our estimator behaves well, even for rare alleles, in the presence of recombination or where data are simulated under a different genealogical process (birth-death model). However, we also quantify (grey lines) the extent of variability in estimates (and uncertainty around them) – which is substantial. Figure 5, which summarises the results on real data, therefore is likely to be a noisy estimate of the truth, hence the various ways we have attempted to summarise both estimates and their uncertainty (note boxes around codons in addition to colours). We apologize for the confusing nature of Figure 5. However, we experimented with many different ways of representing the results and could find nothing clearer. We note that the focus in Figure 5 is on the overlap with known or predicted motifs – hence representation of the selection intensity as coloured bars rather than lines.

As the reviewer points out, the key question is whether we provide additional validation concerning our inferences. This is precisely the motivation behind three analyses presented in the paper. First, we consider a set of known allele-epitope relationships (Figure 4 and Supplementary Text 6) and show that most are recovered in our analyses (with some exceptions that reveal how much uncertainty there can be about what are genuine allele-epitope pairs in experimental data). Second, as summarised in Table 2, we find strong enrichment of experimentally defined HLA epitopes at sites inferred to have high rates of HLA-induced evolution for the corresponding allele (A-list and B-list epitopes). Third, as described in Supplementary Text 10 and summarised in the text (lines 347-354), we find that (at least for HLA-A and HLA-B) closely related alleles induce closely related profiles of selection. We believe that these lines of evidence add credibility to our findings, which, we hope, will stimulate additional experimental analysis of the potentially novel associations described in the manuscript.

A first major aspect that would profit from additional attention is estimation of how related the samples in the database DB are to those in the phenotyped set D. My intuition is that the degree of relatedness is critical to the power of the method and indeed information from strains in D that are closely related to those in DB is much larger than from those with even slightly more distance relations. Specifically, the signal of adaptation that is being searched for is the changes in the most recent infection and therefore the specificity of the signal decreases with each extra transmission between D and DB. This is relevant especially because the degree of genetic relatedness can be estimated from the HMM. It seems to me that the information from strains at different distances to DB could be weighted differently in inference. This factor can also affect the power of inferences and explain why some phenotypes are easier to study than others etc; different phenotype sets can very easily differ systematically in their relatedness to the same database. All of this can be investigated empirically and might possibly explain some of the difference between the real and simulated data? It would also be useful to know for experimental design purposes which strains contributed substantially to the selection signatures. It might even be useful in validation.

In response to these questions, we have carried out an additional series of simulations to investigate the impact of relatedness between D_b and D on the accuracy of inference. These are described in detail in Supplementary Text 4 and summarised in the paper (lines 204-223).

Briefly, we used protease data from Botswana ($n=343$, protease only) to perform a parametric bootstrap, in which data (viral sequences, HLA genotypes) are simulated using inferred parameters. We used the viral sequences from the Botswana study to mimic the infecting viral strains (allowing for recombination), drew individual HLA genotypes at HLA-A, -B and -C by re-sampling with replacement and allowed sequences to evolve stochastically using the inferred parameters. We

then performed inference as in the paper using a variety of strategies that ranged from only using reference viral sequences known to come from outside Botswana (hence most distantly related) to one consisting of the input Botswana strains (hence the best possible data set).

Overall, we found that while using the actual Botswana sequences as reference strains did give the best results (as measured by weighted root mean square error, RMSE, in selection profile, value of 0.766), the differences between strategies were subtle (see Supplementary Figures 9-11). Using the reference data set assembled for the paper (which includes only 79 sequences known to come from Botswana) gave a RMSE of 0.767 and a restricted data set consisting of the 36% of the viral sequences known not to come from Botswana had a RMSE of 0.788. In addition, we found that the strategy of augmenting the reference database with the sequences from query data in a leave-one-out strategy (i.e. all sequences but that under consideration) typically gave worse performance (albeit a minor difference). In short, we conclude that well-matched representation within the reference database (at least by sampling location) is not essential for accurate results.

In terms of measuring the contribution of reference panel strains at different degrees of relatedness, the HMM automatically will weight towards sequences that are closer to that seen in the target individual (and we have an additional step of choosing the 100 closest strains on the basis of Hamming distance to improve computational efficiency – see Methods). Supplementary Figures 3 and 5 show how just using the 10 or 100 closest strains compare – and we only see a marginal gain in accuracy for the larger panel in the presence of recombination, consistent with the notion that closely related sequences will drive inference. It is precisely this idea that motivates the use of the vast reference data set (hence the potential to find at least one strain that is close in sequence to that which infected the individual). What is perhaps surprising is that suitable ancestors can be obtained without dense sampling of the population in question.

I would also like to know what proportion of substitutions inferred by the HMM are likely, under the model, to be explained by selection linked to the measured phenotypes. This might be a useful statistic to output to get an impression of what is going on. Other indication of measured effect sizes would also/alternatively be welcome.

To estimate the fraction of changes driven by HLA-induced selection we introduced an additional summary of the selection pressure estimates, described in Supplementary Text 9. Briefly, the statistic calculates, for each individual, the expected substitutions arising from all forces and those arising from HLA-induced selection where the effect is to increase the rate of escape. Averaged across individuals and sites, we calculate that in protease this represents 18% and 28% of all mutations (59% and 77% of non-synonymous substitutions) away from subtype B and C consensus in protease respectively. In reverse transcriptase, this represented 18% and 32% of all mutations (61% and 78% of non-synonymous substitutions) away from subtype B and C consensus respectively. In short, over half of all protein-coding changes are estimated to have arisen through the result of HLA-induced selection. A summary of these findings is included in lines 341-344 of the main text.

A second major aspect is confounding by population structure. The introduction dwells on this as being an important factor but while the method does seem to provide some protection from the most severe sources of confounding, I do not think it eliminates the risk entirely. However, this is not discussed at all later in the paper. Especially if the focus of the manuscript is new methodology, I think there should be detailed discussion of the issue and also a demonstration of what empirical evidence can be brought to bear to investigate it. Specifically, it seems relevant how the members of D are related to each other and how this compares with their relationships with the strains they are inferred to get ancestry from in DB.

We thank the reviewer for highlighting this and acknowledge that we had previously done a poor job of exploring and explaining robustness to structure. Population structure has several distinct influences on testing for association between viral sequence and host factors.

1. **Sensitivity to confounders.** Differences between populations in HLA allele frequencies and the distribution of viral genomic variation will trivially lead to correlation between viral sequence and HLA alleles when data from multiple studies – as in this work – are combined. We wish our method to be robust to such effects.
2. **Poor calibration of statistical tests arising from relatedness.** The inclusion of multiple sequences that are closely related to each other (but which are treated as if independent) can lead to inflation of test statistics and over-confidence in parameter estimates. We wish our method to provide well-calibrated estimates even in the presence of relatedness.
3. **Population structure driven by evolutionary forces.** Differences in viral sequences between populations may, at least in part, be driven by differences in the genetic composition of hosts (i.e. differences in the frequency of HLA alleles). We wish our method to be able to detect such causal relationships, even in the presence of confounding (as in #1).

The idea of the method presented here is that such population structure is controlled for at the *individual* level, by the automated selection of sequences from the reference panel that act as proxies for the viral sequences that infected the individual. This will potentially break down if there is substantial divergence between the reference sequences and the study samples (e.g. all references were of subtype B and all query sequences were of subtype C). Although such crude stratification can be avoided, the simulations described above using the Botswana data address whether weaker stratification and within-sample relatedness affect the reliability of the estimator. Specifically, we find that excluding Botswana samples from the reference panel (hence exacerbating population structure) leads to only marginally poorer estimates of HLA-associated selection (RMSE of 0.788 compared to 0.766 using ideal reference panel). Importantly, using this reference panel does not lead to additional ‘spikes’ in estimated HLA-induced selection (Supplementary Figure 5). Similarly, because the coverage of the estimator is little affected by the reference panel (frequency-weighted 50% coverage of 57.4% using the ideal panel and 45.9% using the panel that excludes Botswana sequences) we conclude that relatedness among sequences has only a weak impact on calibration.

Using empirical data we can also assess evidence for remaining stratification. In each analysis we identified sites that differ between subtypes B and C – marked on Figures 3, 4, and 5 in the main text. Should stratification be a major problem, then these would be strongly enriched as sites inferred to be under HLA selection, particularly for alleles with big frequency differences between populations. However, most of the positions inferred to be strongly selected by specific HLA alleles are observed at sites that do not differ between subtypes (42/52), and most for one or two HLA alleles. The remaining 10 sites could potentially be driven by stratification, however we note the presence of experimentally characterised epitopes that overlap such sites (see, for example B*35 and B*37; Figure 5), which suggests that differences in HLA allele frequency between populations may be a significant driver of subtype differentiation.

We have added text (lines 204-223) to the paper to describe these analyses and better explain issues around population stratification and how we achieve robustness.

A third more speculative aspect is that it would be interesting to find out more about whether recombination could be related to other evolutionary processes or phenotypes.

We agree that this would be an interesting extension to the work, but feel that it is beyond the scope of the current paper. We now note (in the discussion – lines 394-395) the possibility of extending the method to consider association between covariates and parameters other than selection coefficients.

Reviewer 2

Palmer and colleagues describe a joint statistical model for simultaneously estimating parameterized escape (e.g. from HLA or antiretrovirals), reversion, and selection from unknown sources while correcting for population structure and recombination. The key observation is that the existence of massive reference datasets allow for some simplifying assumptions that make such estimation more tractable than possible with smaller datasets. The model described is elegant, the experiments generally well done, and the manuscript uncommonly well written. While I do have some comments on how the manuscript should be improved, overall it is an important and useful contribution.

We thank the reviewer for their supportive comments.

The manuscript is logically broken into three sections, and I will comment on them as such.

1) The model. Overall, the model is well described and mostly well motivated. I do have some modest concerns however.

a. I am unconvinced of the motivation for the block prior. In the case of ARV, is there evidence that we should expect selection from a specific drug to uniformly impact a window of a protein? In the case of HLA, an epitope provides such a window, but most of the available evidence appears to suggest that actual selection occurs at a small number of discrete sites within the epitope. The implementation of the window is a bit confusing to follow, and I expect computationally expensive while possibly increasing the variance on parameter estimates. Is it justified?

The piece-wise constant prior used in the modelling was motivated by the block-like nature of epitopes, though in fact can accommodate a wide range of selection landscapes (including isolated sites of selection) due to its probabilistic nature. We chose the prior because it effectively reduces the number of parameters fitted (for example, compared to a model in which every codon had a separate value). The computational cost of the method is relatively low (due to efficient proposal moves within the MCMC) and the simulations and analysis of drug resistance demonstrate that the approach can handle a wide range of biological landscapes. We note that a model of independence between sites can be achieved by setting the expected block size to be small.

Within the manuscript we have now included additional text to explain our choice of the piecewise constant prior (lines 135-138) and additional text within the supplement.

b. The sheer number of parameters estimated is large and I worry about overfitting. While the datasets are very large, when considering, e.g., HLA alleles that occur in 10 individuals (their lower limit in the real data), the effective size for parameter estimation is very small. Is this a concern?

The size and complexity of the model we are trying to fit is precisely why we have chosen shrinkage priors that – in the absence of signal – will pull estimates towards the null. Likewise, the piecewise constant prior has a similar effect (encouraging nearby sites to have similar parameter values unless the data are sufficiently strong to indicate otherwise). The simulations we performed were chosen to be sufficiently realistic in terms of profile complexity and allele frequency to assess how the estimator performs. As shown in Figure 2 and Supplementary Figures 2 and 4, while there is noise in parameter estimates, we have excellent sensitivity and specificity for ten-fold HLA-induced changes of substitution rate for both common and rare alleles. Weaker effects will be harder to detect for rare alleles. Importantly, where there are no effects, the estimator performs well in terms of mean estimate and coverage.

c. The authors had seemingly small comments about truncating their data that were not discussed. E.g., using 2 digit HLA resolution (when available evidence suggest a majority of escape is specific to 4 digit alleles), or conversely not grouping classes of drugs. Would their model be amenable to a hierarchical prior, which has the potential to automatically select the level of grouping most appropriate for a given codon? A hierarchical prior could also provide shrinkage to deal with overfitting.

We agree with the reviewer that analysis at 4-digit resolution – or with a hierarchical prior – would be preferable, though additional work is needed to implement such a model. We justify the choice of 2-digit in three ways. First, 27% of our data is only typed to 2-digit resolution. Second, at 4-digit resolution 60% of HLA alleles have <10 copies in the dataset (compared to 30% at 2-digit resolution), so we would lose considerable power (unless 4-digit alleles have effectively independent selection profiles). Thirdly, while different 4-digit alleles will have some different binding properties, often they are very similar to each other. Indeed, as shown in this paper (Supplementary Text 11), even at 2-digit level we find that related HLA alleles have related selection profiles.

We have included additional text (lines 278-281) to summarise the motivation for the 2-digit analysis level and now mention extension to hierarchical modelling within the discussion (lines 394-395).

d. The separation of D from Db was not well motivated. One could easily construct cases where the majority of samples from a given region are in the query set, not reference. E.g., in their HLA data, I imagine the majority of available samples from the city of Durban are in their query set, which would violate their assumption that the reference set is an effective estimate of the population of available sequences at time of transmission. But could not D be included in Db (with the constraint that a specific sequence *i* cannot be derived from itself)?

Following the points raised by both reviewers, we have carried out additional analyses to explore the effect of population differentiation between the reference and query data sets and to explore the value of leave-one-out (LOO) analyses that augment D_B with samples from D (minus the sample being analysed). These are described above, in Supplementary Text 4, and also summarised in the manuscript (lines 204-223) and Supplementary Figures 9-11.

Briefly, we carried a series of analyses using data from Botswana ($n=343$, protease only) to simulate new data (viral sequences and host genotypes) using the HLA allele frequencies and evolutionary parameters estimated in the study. We then analysed the simulated data using a series of reference panels ranging from the perfect set (the sequences used as ancestors in the simulation) to one that excluded all data from Botswana (hence strongly differentiated). We also considered, for each panel, the value of the LOO approach. We measured performance by the root mean square error (RMSE) between the true and inferred HLA-allele specific selection profiles (weighted by HLA allele frequency). Overall, we found relatively little difference in performance between the best and worst reference panels (RMSE of 1.030 v 1.059, an increase of 3%). The LOO strategy only helped in settings where the reference panel was of comparable size to the data set being analysed. Indeed, often the approach has slightly worse performance, presumably due to independence assumption breaking down when sequences in D are used as references.

Because of the value of the LOO approach under certain conditions we have made this option available to users. In the Github code, this is included as the `-q` or `--separate_reference_fasta` options. In the supplement, we have added the following sentence:

“Users of our software (available at <https://github.com/astheegggs/mcqueen>) can set D_B to be created using a leave one out approach as in a) and b) by using the $-q$ or $--separate_reference_fasta$ flags.”

e. The parameterization of reversion appears to be underdetermined. If I'm following the model correctly, reversion is “independent of HLA” (shouldn't it be only among individuals without a given HLA?), then multiplied into ω_i (S36). But ω and ζ are estimated independently for each site. How is ω_i and ζ_i then identified? Does this explain why reversion is not well estimated in the synthetic data?

We thank the reviewer for pointing out the lack of clarity in the text. The reversion rate is ζ_i is present in all individuals. We imagine ζ_i to reflect the selection towards the consensus strain at site i . HLA associated selection is superimposed on top of this to select away from the consensus codon. ω_i is the dN/dS ratio at site i for any non-synonymous single codon, whereas ζ_i only multiplies ω_i for non-synonymous base changes from consensus to non-consensus at site i . This distinction makes the problem identifiable. In the main text, we have added (lines 458-459):

“Note that reversion only acts on non-synonymous changes toward the consensus codon C , and as such the parameterisation is identifiable.”

In the supplement, we have added:

“Note that reversion only acts on non-synonymous changes that result in the consensus codon C as such the parameterisation is identifiable and we are able to distinguish ω_i and ζ_i in the product $\omega_i \zeta_i$ ”

f. Please double check S19. Transition and transversion should be swapped for the last two.

This was an error and has been corrected.

2) Synthetic experiments

a. The authors should explore what happens when their assumptions are not met, as will undoubtedly be the case to some degree. E.g., what happens if data are generated from the Botswana sequences, then all Botswana sequences are removed from Db (simulating the case where the query sequences make up the majority of known sequences for a region)? How massive does Db need to be? They ran their Birth-Death model sampling 10% = 100k sequences. What if they had 10k? 1k? The question is, is HIV Pr/RT and Influenza H/N the only datasets we can use this for, or will it work for other viruses (or proteins) with less data? When there is not enough Db data, or when the data in Db is not similar enough to D, does the model fail gracefully?

We have addressed these concerns by sub-sampling 10% and 1% of the reference data in Simulation Study 2 (lines 195-203 Supplementary Figure 8 and Supplementary Text 3). In addition, the synthetic experiments using the Botswana data (described above) also examine the impact of changing the reference panel size and composition. We find that with 10k samples, in the Birth-Death model simulation, sensitivity is little affected (0.49 at FPR = 0.01, compared to 0.65 with 100k samples). With 1k samples (1%), sensitivity is further reduced (sensitivity = 0.32). The LOO approach only adds value when the reference data set is of comparable size to the query set (sensitivity after adding the query sequences to the 1k reference data set increased to 0.38). However, in the extreme case of no reference data, the LOO approach performs well (sensitivity = 0.34). As described above, we now include a LOO strategy as an option in the software.

b. Were the tests of “common” vs “rare” alleles independent, and containing a single HLA? Is this overly optimistic compared to including a realistic number of Hla alleles? It seems it may be easier for the model to converge if the number of alleles is small.

All simulations carried out involved joint analysis of multiple alleles. This was not clear in the paper. We now include the statement (lines 154-156).

“We performed joint inference on the HLA associated selection across the viral sequence and examined how the size of the sample-specific reference panel (range 10 to 100) affects estimates.”

c. In terms of comparisons, I would expect that the present model will benefit most compared to other models when the query size $|D|$ is small, as this model can leverage D_b ? E.g., in the Real HLA set considered, 3k sequences in D is enough that I imagine most of the models will have quite a bit of statistical power.

To address this question, we repeated the comparison between methods using a larger data set of 3000 query sequences (i.e. D). The results are shown in Supplementary Figure 7 and summarised in the text (lines 188-191). We found that the model-based approach retains a substantial power in sensitivity over alternative methods (sensitivity of 0.81 at an FPR of 0.01, compared to the next best-performing method, PhyloD, at 0.22). The gain in power through the method presented here results both from the reference data set size but also the inherent shrinkage within the modelling approach.

3) Real data

a. What were inferred recombination rates? Is there evidence this is an important feature of the model?

The estimated recombination rates are low (now shown in Supplementary Figure 19). Given the estimated parameters, we expect 7.3% of protease and 16.8% of the analysed portion of reverse-transcriptase sequences to have undergone at least one recombination event. The following has been added to the main text (lines 395-398).

“Importantly, within the data analysed here, we estimate that between 7% (protease) and 17% (RT) of viral samples having undergone recombination at least once since their common ancestor with sequences in the reference data set.”

b. I do not follow the motivation for adding 2k sequences from D_b to D for the drug association study.

We apologise for this being unclear. The rationale behind adding 2000 sequences from D_b to D is to ensure that there is variability among sequence within D in drug selection. If, for example, all protease inhibitors had a substantial, but identical selective impact, then there would be no variability within D , hence we could not estimate the selection pressure. By including 2000 sequences from patients that have not – to our knowledge – had protease inhibitor treatment we include a ‘null’ set that guarantees variability. In reality this step may not have been necessary. We have summarised this justification in the paper (lines 233-235):

To guarantee the presence of variation in drug selection we introduce a null class by randomly assigning 2000 sequences from D_b to D .

c. Why are HLA A-, B- and C- treated independently? There was nothing in the model description that suggests to me these can't be jointly estimated, unless it's a matter of reducing the number of parameters? (in which case, a hierarchical prior might help)

There was an error in the original submission. All analyses have been carried out jointly. We have now made this clear throughout.

d. The authors should probably not claim this to be “the most complete and comprehensive to date...covering nearly 3,000” HLA-linked Pr and RT sequences. E.g., Carlson (and some of the present authors) 2016 (PMID 27183217) included over 3k HLA-linked sequences from Gag, Pol and Nef. Comparing the results might be an interesting exercise, especially considering the subtype C sequences for the two studies largely overlap.

We have removed this statement. We have also included a comparison of results, paying particular attention to inference selection away from the subtype C viral reference. We find that the intersection of the sites found in our top-tier sites and the Carlson *et al* set of $q < 0.2$ association lie exclusively in well defined A-list and B-list epitopes for HLA-B alleles. Subsets of the remaining associations lie in well defined epitopes suggesting that the methods are complementary and detecting distinct components of the remaining signal. We summarise these findings in the manuscript (lines 313-319) and in Supplementary Tables 6-7.

e. In the discussion, the authors note that some HLA escapes are at sites of subtype consensus differences. This observation has previously been made for Gag, where it was further noted that HLAs common to subtype C (e.g.) were selecting for the C consensus (Carlson et al, 2008; PMID 19023406). Do the authors similarly find sites where alleles are associated with selection of consensus, which I think would manifest as $\gamma \ll 1$?

Motivated by this question, we have carried out a more substantial analysis of the impact of population differences in HLA allele frequency on selection to and away from consensus sequences for subtypes B and C. The results are summarised in Supplementary Text 10, Supplementary Table 8 and Supplementary Figure 20.

Briefly, we used the inferred HLA selection profiles away from the B and C consensus sequences, combined with HLA genotypes for individuals inferred to have B or C HIV sequences. We then used permutation analysis to ask whether individuals with subtype B sequences tend to select away from subtype C more than away from subtype B (and vice versa). We found a signal of selection by HLA alleles common in hosts of subtype C viruses away from subtype B consensus in protease ($p = 2.8 \times 10^{-5}$) and reverse transcriptase ($p = 2.0 \times 10^{-6}$). We also observe a signal of selection away from subtype C consensus by HLA alleles common in hosts of subtype B viruses in reverse transcriptase ($p = 0.0058$), but we did not observe an effect in protease ($p = 0.70$). The effects are, however, relatively weak (Supplementary Figure 20).

In contrast, we did not observe selection towards subtype consensus by HLA types (which would indeed manifest as $\gamma \ll 1$). Sites that differ between subtype B and C (in both RT and protease) appear to tolerate mutation more than average in both subtype B host populations and subtype C host populations (i.e. the red points in Supplementary Figure 20 tend to be in the top right corner of the plot).

f. It might be noted that including the early infection SPARTAC cohort may be reducing their power, as these individuals may not have been infected long enough for escape to have occurred, thereby diluting the escape signal (e.g. Martin et al, 2014, PMID 25212686).

This is an interesting point, though we feel the increased sample size gained by inclusion of the SPARTAC cohort outweighs any issues with sampling design.

REVIEWERS' COMMENTS:

Reviewer #1 (Remarks to the Author):

I thank the authors for their extensive revisions to the manuscript and detailed rebuttal letter. The changes are very appropriate and enhance the manuscript. I look forward to seeing it in print.

Reviewer #2 (Remarks to the Author):

The authors have carefully and thoughtfully addressed all of my major concerns. I am pleased to recommend publication of the manuscript.

Response to Reviewers

Reviewer 1

I thank the authors for their extensive revisions to the manuscript and detailed rebuttal letter. The changes are very appropriate and enhance the manuscript. I look forward to seeing it in print.

We thank the reviewers for their kind comments.

Reviewer 2

The authors have carefully and thoughtfully addressed all of my major concerns. I am pleased to recommend publication of the manuscript.

We thank the reviewers for their kind comments.